# Coarse-to-Fine Learning of Dynamic Causal Structures

**Dezhi Yang**
School of Software
Shandong University
Jinan, China
`dzyang@mail.sdu.edu.cn`

**Qiaoyu Tan**
Department of Computer Science
New York University
Shanghai, China
`qiaoyu.tan@nyu.edu`

**Carlotta Domeniconi**
Department of Computer Science
George Mason University
VA, USA
`carlotta@cs.gmu.edu`

**Jun Wang**
Shandong Institutes of Industrial Technology
Jinan, China
`kingjun@sdu.edu.cn`

**Lizhen Cui, Guoxian Yu**[*]
School of Software
Shandong University
Jinan, China
`{clz, gxyu}@sdu.edu.cn`

## Abstract

Learning the dynamic causal structure of time series is a challenging problem. Most existing approaches rely on distributional or structural invariance to uncover underlying causal dynamics, assuming stationary or partially stationary causality. However, these assumptions often conflict with the complex, time-varying causal relationships observed in real-world systems. This motivates the need for methods that address *fully dynamic causality*, where both instantaneous and lagged dependencies evolve over time. Such a setting poses significant challenges for the efficiency and stability of causal discovery. To address these challenges, we introduce `DyCausal`, a dynamic causal structure learning framework. `DyCausal` leverages convolutional networks to capture causal patterns within coarse-grained time windows, and then applies linear interpolation to refine causal structures at each time step, thereby recovering fine-grained and time-varying causal graphs. In addition, we propose an acyclic constraint based on matrix norm scaling, which improves efficiency while effectively constraining loops in evolving causal structures. Comprehensive evaluations on both synthetic and real-world datasets demonstrate that `DyCausal` achieves superior performance compared to existing methods, offering a stable and efficient approach for identifying fully dynamic causal structures from coarse to fine.

## 1 Introduction

Temporal causal relationships can decipher the dynamics of complex real-world processes and play critical roles in boosting scientific discoveries in a wide range of disciplines, such as traffic (Cheng et al., 2024b), healthcare (Qian et al., 2020), meteorology (Nowack et al., 2020), and finance (Hammoudeh et al., 2020). Granger causality (Granger, 1969) serves as the cornerstone for temporal causal analysis; it determines causality based on the predictability of the cause series to the effect series. Its high explainability and compatibility with deep models continue to attract research dedicated to uncovering causal graphs from observed time series (Marinazzo et al., 2008; Khanna & Tan, 2019; Liu et al., 2024).

---

[*]Corresponding author

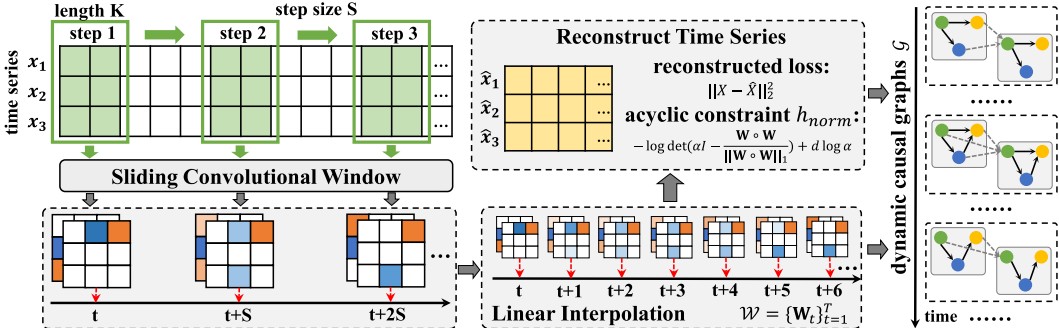

Figure 1: Conceptual overview of `DyCausal`. `DyCausal` uses a sliding convolutional window to traverse temporal series and encodes coarse-grained time-varying causal structures. It then defines a linear interpolation strategy inspired by Taylor expansion to refine the causal structures at each time step $\{\mathbf{W}_t\}_{t=1}^{T}$. Based on the acyclic constraint $h_{norm}$ enforced by matrix 1-norm, `DyCausal` reconstructs time series to optimize causal structures and obtains dynamic causal graphs $\mathcal{G}$ that conform to the dynamic distribution of time series.

Recognizing and modeling dynamic causality is crucial for unraveling the underlying causal mechanisms in complex systems. Despite substantial progress in temporal causal discovery (Runge et al., 2019; Pamfil et al., 2020; De Brouwer et al., 2020; Bellot et al., 2022; Tank et al., 2021; Sun et al., 2023; Zhou et al., 2024; Wu et al., 2024; Yao et al., 2025), these methods still focus on evolving time series with static causal models, and thus cannot fulfill the demand of identifying dynamic causality in real-world applications. In dynamic causal settings, time series often exhibit changing distributions, with causal relationships emerging, changing, or vanishing over time (e.g., seasonal shifts in patient disease trajectories). To handle this, some recent methods (Fujiwara et al., 2023; Cheng et al., 2025) impose static assumptions on certain causal components to model partial dynamic causality. This often runs counter to the inherent dynamic nature of the real world. How to realize causal discovery towards more practical applications, faithfully modeling fully dynamic causality, where all causal relations can change over time, is urgently necessary but has not been well explored yet.

Learning *fully dynamic* temporal causal structures is quite challenging, primarily due to two key obstacles: (i) **efficient and stable discovery of time-varying causal graphs**, given the complexity of dynamic signals; (ii) **effective enforcement of acyclicity within causal graphs**, especially when causal relationships are rapidly changing. The frequently changing causality at each time step causes expensive and hard-to-converge causal models. Fortunately, for causality that changes continuously over time, a good identification of its coarse-grained states at a few time steps can, in principle, support approximating its complete time-varying trajectory. To this end, we propose a framework for coarse-to-fine learning fully dynamic causal structures (`DyCausal`). Specifically, `DyCausal` leverages convolutional networks to encode causal structures within coarse-grained sliding windows, and then refines causal structures at each time step through linear interpolation. In addition, an acyclic constraint based on matrix norm scaling is introduced to optimize the causal model. By scaling the causal matrix to an optimizable space, the constraint guides the model to identify causal graphs with improved stability and efficiency. Figure 1 illustrates the conceptual framework of `DyCausal`. Our main contributions are summarized as follows:

(i) `DyCausal` meticulously combines the sliding convolutional window and linear interpolation to efficiently capture the coarse-to-fine temporal evolution of causal structures. This design successfully models fully dynamic causal graphs and boosts flexible adaption to various causal models, broadening its applicability to diverse domains.

(ii) `DyCausal` defines an enhanced strategy for the log-determinant-based acyclic constraint, utilizing the matrix norm to achieve an always differentiable and optimizable constraint. This strategy not only avoids retraining and hyperparameter search, but also improves the stability and efficiency of learning dynamic causal graphs.

(iii) Extensive experiments on both synthetic and real-world datasets show that `DyCausal` consistently outperforms state-of-the-art methods, demonstrating its ability to capture fully dynamic causal evolution from coarse to fine while enforcing a theoretically stable acyclic constraint.

## 2 RELATED WORK

Identifying causal graphs from time series has become an important topic in causal learning, with existing approaches broadly falling into two categories: constraint-based and score-based methods. Constraint-based methods (Entner & Hoyer, 2010) extend the two classic algorithms, PC (Spirtes & Glymour, 1991) and FCI (Spirtes et al., 2000), to time series and determine causal relationships by conditional independence tests. Runge et al. (2019); Gerhardus & Runge (2020) later combined these two algorithms with linear and nonlinear conditional independence tests and achieved scalability on large-scale time series. Score-based methods (Hyvärinen et al., 2010; 2008; Tank et al., 2021) use vector autoregressive loss and sparse regularization to identify causal models that correctly reconstruct time series. Inspired by NOTEARS (Zheng et al., 2018), Pamfil et al. (2020); Sun et al. (2023) introduced the differentiable acyclic constraint to optimize autoregressive processes in linear and nonlinear models. Gao et al. (2022); Li et al. (2023) considered intervention data to enhance the identifiability of temporal causal structures. In addition, Bellot et al. (2022) used the ordinary differential equation to reconstruct time series. Many studies (Khanna & Tan, 2019; De Brouwer et al., 2020; Cheng et al., 2023; 2024a), including the aforementioned ones, have made great progress, but all assume that the time series evolve from *static* causality and fail to identify *dynamic* causality as shown in the experimental section, which limits their application in the real world.

We focus on more realistic scenarios with dynamic causal relationships, causal discovery for such scenarios is still under-explored. In this context, Willig et al. (2025) introduced a meta-causal model to capture dynamic causal relationships in the bivariate system. Saggioro et al. (2020); Gao et al. (2023) divided time series into segments and identified the causal mechanism in each segment. Gao et al. (2023) modeled the causality as a smooth function of the time index. These methods focus on the dynamic data distribution, but are still limited to learning static causal structures. Song et al. (2009); Gao & Yang (2022) utilized kernel-reweighted autoregression to model dynamic causal graphs. Hallac et al. (2017) developed a message-passing algorithm to infer time-varying structures. Cheng et al. (2025) used the ordinary differential equation to model the latent dynamics of causal structures. These methods introduce the dynamic to lagged or instantaneous structures only, and *none of them learns the fully dynamic causality*. In contrast, our `DyCausal` leverages sliding convolutional windows and linear interpolation to recover fully dynamic causality from coarse to fine, and a constraint based on the matrix norm to enforce acyclic causal graphs with improved efficiency and stability.

## 3 PROBLEM DEFINITION

Given $N$ independent time series $\mathcal{X} = \{\mathbf{X}_{1:T}^1, \cdots, \mathbf{X}_{1:T}^N\}$ of length $T$, the series $n \in [1, N]$ at time $t \in [1, T]$ records $d$ variable values $\mathbf{X}_t^n = \{x_{t,1}^n, \cdots, x_{t,d}^n\}$. With a slight abuse of notation, we denote the observed data at time $t$ as $\mathbf{X}_t$ and ignore the serial index. Without loss of generality, we assume that all series have the same time-varying causal structures.

**Structural equation model.** We introduce the structural equation model (SEM) to formalize temporal causality, where the generation process of $x_{t,i}$ is defined as follows:

$$x_{t,i} = f_i(\{PA_{ins}(x_{t,i}), PA_{lag}(x_{t,i})\}) + \epsilon_i \tag{1}$$

where $\epsilon_i$ is the exogenous noise independent of the observed variables, $PA_{ins}(x_{t,i})$ and $PA_{lag}(x_{t,i})$ respectively represent the parents that have instantaneous and lagged effects on $x_{t,i}$. In general, SEM assumes a maximum lag $\tau$.

Given generation functions $\mathcal{F} = \{f_1, \cdots, f_d\}$, we denote the causal structure as a weighted adjacency matrix composed of partial derivatives $\mathbf{W}(\mathcal{F}) = [\partial f_1|\cdots|\partial f_d] \in \mathbb{R}^{d\tau \times d}$. $\mathbf{W}[j + pd, i] = 0$, $p \in [0, \tau]$ if and only if $x_{t-p,j}$ is independent on $x_{t,i}$.

**Acyclicity.** We assume that there is no cycle in instantaneous causal relationships. This is a key assumption in causal discovery, simplifying the difficulty of modeling causality and ensuring distinguishable causes and effects. Unlike non-temporal causal discovery, temporal causal graphs contain both instantaneous and lagged causal relationships, where the causes of lagged causality clearly occur before the effects and do not involve acyclicity. By splitting the causal matrix $\mathbf{W}$ into the instantaneous matrix $\mathbf{W}^{ins}$ and lagged one $\mathbf{W}^{lag}$, we only require that $\mathbf{W}^{ins}$ be a directed acyclic graph, $\mathbf{W}^{ins} \in DAGs$.

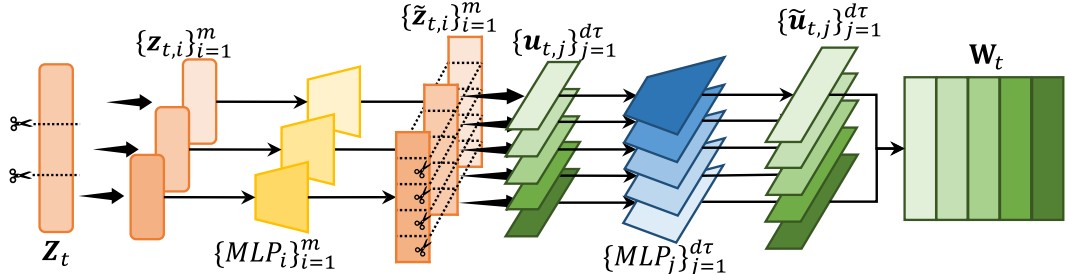

Figure 2: The decoding process. Vector $\mathbf{Z}_t$ is cut and decoded in parallel to obtain vectors $\{\tilde{\mathbf{z}}_{t,i}\}_{i=1}^{m}$, which are again cut, recombined and decoded in parallel to obtain vectors $\{\tilde{\mathbf{u}}_{t,j} \in \mathbb{R}^{dm}\}_{j=1}^{d\tau}$. The combination of $\{\tilde{\mathbf{u}}_{t,j} \in \mathbb{R}^{dm}\}_{j=1}^{d\tau}$ forms the matrix $\mathbf{W}_t$.

**Dynamic causal structures.** Given the ground-truth weighted adjacency matrix $\mathbf{W}_t$, $t \in [1, T]$, we denote its time-varying trajectory as a smooth function of the time index as follows:

$$\mathbf{W}_t = \mathbf{F}(\mathbf{W}_0, t), \mathbf{W}_t^{ins} \in DAGs \tag{2}$$

In this paper, we are committed to learning the time-varying functions $\mathbf{F}$ and extracting the weighted adjacency matrix $\mathbf{W}_t$.

## 4 FULLY DYNAMIC CAUSAL STRUCTURE LEARNING

In this section, we introduce a coarse-to-fine strategy to encode the time-varying trajectory of causal structures, and propose an efficient acyclic constraint integrated into the optimization process, resulting in a reliable framework for identifying fully dynamic causal graphs.

### 4.1 COARSE-TO-FINE ENCODING OF CAUSAL MATRIX

We propose to mine the *coarse-grained states* of dynamic causal structures across multiple time steps. Referring to Eq. (2), the causality is defined as continuously evolving over time, and the few coarse-grained states of dynamic causality can support approximating its fine-grained time-varying trajectory. We design a sliding convolutional window of width $N$ and length $K$, and slide it over the time series with step size $S$. At each sliding step, the convolutional network encodes the data within the window as a state vector:

$$\mathbf{Z}_t = CNN(\mathbf{X}_{t:t+K}) \tag{3}$$

where $\mathbf{X}_{t:t+K}$ are the data within the window, $CNN(\cdot)$ is the convolutional neural network, and $\mathbf{Z}_t \in \mathbb{R}^{dm}$ embeds the data as a coarse-grained state representation of causal structures. We defer the discussion of $m$ in Section 4.3. Next, we decode $\mathbf{Z}_t$ onto a weight adjacency matrix to represent causal structures at time $t$.

We introduce a *parallel decoding strategy* to reduce parameters and computations (Fig. 2). We divide $\mathbf{Z}_t$ into $m$ short vectors $\mathbf{Z}_t \rightarrow \{\mathbf{z}_{t,i} \in \mathbb{R}^d\}_{i=1}^{m}$, and decode them as $\{\tilde{\mathbf{z}}_{t,i} \in \mathbb{R}^{d\tau}\}_{i=1}^{m}$ using $m$ parallel multilayer perceptrons. We then recombine the elements with the same position in vectors $\{\tilde{\mathbf{z}}_{t,i}\}_{i=1}^{m}$ into $d\tau$ vectors $\{\mathbf{u}_{t,j} \in \mathbb{R}^m\}_{j=1}^{d\tau}$, and further decode them as $\{\tilde{\mathbf{u}}_{t,j} \in \mathbb{R}^{dm}\}_{j=1}^{d\tau}$ with $d\tau$ parallel multilayer perceptrons. Finally, we obtain the causal matrix $\mathbf{W}_t \in \mathbb{R}^{dm \times d\tau}$:

$$\mathbf{W}_t = [\tilde{\mathbf{u}}_{t,1}|\cdots|\tilde{\mathbf{u}}_{t,d\tau}], \; \tilde{\mathbf{u}}_{t,j} = MLP_j(\sigma(\mathbf{u}_{t,j}))$$

$$\{\mathbf{u}_{t,j}|j \in [1, d\tau]\} \xleftarrow{\text{recombine}} \{\tilde{\mathbf{z}}_{t,i}|i \in [1, m]\}, \; \tilde{\mathbf{z}}_{t,i} = MLP_i(\sigma(\mathbf{z}_{t,i})) \tag{4}$$

where $MLP_j(\cdot)$ denotes the multilayer perceptron and $\sigma(\cdot)$ is an activation function. Compared with direct decoding, our strategy reduces the number of parameters from $(d^3m^2\tau)$ to $(d^2m\tau + d^2m^2\tau)$ and the computational complexity from $O(d^3m^2\tau)$ to $O(d^2m^2\tau)$.

The causal matrices obtained above are separated by $S$ time steps and thus remain in coarse-grained dynamics. We approximate their *fine-grained piecewise linear trajectory* by supplementing the

causal matrix at each time step with linear interpolation. To this end, we further assume that $S$ is small enough based on Eq. (2), and the variation of the causal matrix in the interval $[t, t+S]$ can be modeled as a deterministic linear function by the first-order Taylor formula:

$$\mathbf{W}_{t+s} = \mathbf{W}_t + \frac{(\mathbf{W}_{t+S} - \mathbf{W}_t)}{S} \cdot s, \ s \in [0, S] \tag{5}$$

The above formula is equivalent to linear interpolation, and we estimate the causal matrix at each time step, while only calculating $\lfloor (T-K)/S \rfloor$ matrices in Eq. (4). Although a smooth time-varying trajectory of causal structures requires a small $S$, we can easily cut the computation by more than half with $S \geq 2$.

## 4.2 MODIFICATION OF THE ACYCLIC CONSTRAINT

The goal of an acyclic constraint is to prevent variables from self-reconstruction to identify reliable causal structures. Existing acyclic constraints ($h_{exp} = \mathrm{Tr}(e^{\mathbf{W} \circ \mathbf{W}}) - d$) (Zheng et al., 2018) and ($h_{poly} = \mathrm{Tr}((\mathbf{I} - \frac{1}{d}\mathbf{W} \circ \mathbf{W})^d) - d$) (Yu et al., 2019) rely on the trace of the matrix to measure the cyclic characteristics. Subsequent studies (Zhu et al., 2020; Lachapelle et al., 2020; Zheng et al., 2020b; Kyono et al., 2020) basically follow these two constraints, learning causal structures with the augmented Lagrangian format and the L-BFGS-B algorithm (Nocedal & Wright, 2006), and result in unacceptable inefficiency. Although the log-det acyclic constraint ($h_{log} = -\log \det(\alpha I - \mathbf{W} \circ \mathbf{W}) + d \log \alpha$) (Bello et al., 2022) improves efficiency, it remains in a finite optimizable space $\mathcal{W}^s = \{\mathbf{W} \in \mathbb{R}^{d \times d} | \rho(\mathbf{W} \circ \mathbf{W}) < \alpha\}$. Once $\rho(\mathbf{W} \circ \mathbf{W}) \geq \alpha$, the model is forced to reduce the learning rate and retrain, this severely undermines the efficiency and stability of causal discovery.

We propose an efficient and effective log-det acyclic constraint. It is well known that the norm of a matrix is larger than (or equal to) its spectral radius. Thus, we suggest scaling $\mathbf{W} \circ \mathbf{W}$ by its 1-norm to keep it always differentiable and optimizable, giving the following new acyclic constraint:

$$h_{norm} = -\log \det(\alpha I - \frac{\mathbf{W} \circ \mathbf{W}}{||\mathbf{W} \circ \mathbf{W}||_1}) + d \log \alpha \tag{6}$$

In this constraint, $\alpha$ is a constant(close to 1 but strictly larger, e.g. 1.001, defined as $1^+$) and does not need additional adjustments, and $||\mathbf{W} \circ \mathbf{W}||_1$ is the 1-norm of $\mathbf{W} \circ \mathbf{W}$. The scaled matrix is always within the optimizable space ($\rho(\frac{\mathbf{W} \circ \mathbf{W}}{||\mathbf{W} \circ \mathbf{W}||_1}) < 1^+$) and avoids redundant retraining and hyperparameter search. Although our constraint penalizes the cyclicity of $\frac{\mathbf{W} \circ \mathbf{W}}{||\mathbf{W} \circ \mathbf{W}||_1}$, the cyclic characteristic of $\frac{\mathbf{W} \circ \mathbf{W}}{||\mathbf{W} \circ \mathbf{W}||_1}$ is consistent with that of $\mathbf{W} \circ \mathbf{W}$, and $\frac{\mathbf{W} \circ \mathbf{W}}{||\mathbf{W} \circ \mathbf{W}||_1}$ is a DAG if and only if $\mathbf{W} \circ \mathbf{W}$ is a DAG. In addition, $\nabla h_{norm} = \frac{2}{||\mathbf{W} \circ \mathbf{W}||_1}(\alpha I - \frac{\mathbf{W} \circ \mathbf{W}}{||\mathbf{W} \circ \mathbf{W}||_1})^{-\top} \circ \mathbf{W}$ and $\nabla h_{log} = 2(\alpha I - \mathbf{W} \circ \mathbf{W})^{-\top} \circ \mathbf{W}$ also have the same characteristic, and $\nabla h_{norm} = 0$ if and only if $\nabla h_{log} = 0$.

**Theorem 1.** *Characterization. Given the acyclic constraint* $h_{norm} = -\log \det(\alpha I - \frac{\mathbf{W} \circ \mathbf{W}}{||\mathbf{W} \circ \mathbf{W}||_1}) + d \log \alpha$. *Then, the following holds:*

    (i) $h_{norm} \geq 0$, *with* $h_{norm} = 0$ *if and only if* $\mathbf{W} \in DAGs$.

    (ii) $\nabla h_{norm} = \frac{2}{||\mathbf{W} \circ \mathbf{W}||_1}(\alpha I - \frac{\mathbf{W} \circ \mathbf{W}}{||\mathbf{W} \circ \mathbf{W}||_1})^{-\top} \circ \mathbf{W}$, *with* $\Delta h_{norm} = 0$ *if and only if* $\mathbf{W} \in DAGs$.

It follows from Theorem 1 that $h_{norm}$ inherits the excellent characteristics of $h_{log}$. Referring to Definition 3.3 in Nazaret et al. (2024), our acyclic constraint satisfies three stability criteria:

**Theorem 2.** *Stability. The acyclic constraint* $h_{norm}$ *is stable. For almost every* $\mathbf{A} = \frac{\mathbf{W} \circ \mathbf{W}}{||\mathbf{W} \circ \mathbf{W}||_1} \in \mathbb{R}_{\geq 0}^{d \times d}$:

    (i) **E-stable** $h_{norm}(k\mathbf{A}) = \mathcal{O}_{k \to +\infty}(1)$.

    (ii) **V-stable** $h_{norm}(\mathbf{A}) \neq 0 \Rightarrow h_{norm}(k\mathbf{A}) = \Omega_{k \to 0^+}(1)$.

    (iii) **D-stable** $h_{norm}$ *and* $\nabla h_{norm}$ *are defined almost everywhere.*

*E-stable* and *V-stable* ensure that $h_{norm}$ neither explodes to infinity nor rapidly vanishes to 0. *D-stable* means that the acyclic constraint and its gradient are always computable for almost every $\mathbf{A}$.

---

**Algorithm 1** `DyCausal`: Coarse-to-fine learning of dynamic causal structures

---

**Input**: Cause series $\mathcal{X}^c$, effect series $\mathcal{Y}$, penalty coefficient $\beta$, decay coefficient $\mu$ (default is 1), decay rate $\gamma$ (default is 0.1), number of iterations $r$ (default is 4) and threshold $\delta$

**Output**: Dynamic causal structures $\mathcal{G} = \{\mathbf{G}_t\}_{t=\tau+1}^{T}$

1: Initialize parameters $\theta$
2: **for** $epoch = 1$ to $r$ **do**
3:     Reconstruct effect series: $\hat{\mathbf{Y}}_{t:t+K} = \mathbf{W}_t \mathbf{X}_{t:t+K}^c$
4:     Minimize the objective: $\min \mu(\frac{1}{NTK} \sum_{t=\tau+1}^{T-K} ||\mathbf{Y}_{t:t+K} - \hat{\mathbf{Y}}_{t:t+K}||_2^2 + \beta|\mathbf{W}_t|_1) + h_{norm}$
5:     Update coefficient $\mu$: $\mu = \gamma\mu$
6: **end for**
7: Prune $\mathcal{W} = \{\mathbf{W}_t\}_{t=\tau+1}^{T}$: $\mathcal{G} = \{\mathbf{W}_t > \delta\}_{t=\tau+1}^{T}$
8: **return** $\mathcal{G}$

---

In particular, $\nabla \rho(\mathbf{A})$ may not be calculable if $\mathbf{A}$ has repeated eigenvalues or if $\rho(\mathbf{A})$ corresponds to multiple eigenvalues. Therefore, $h_{norm}$ constrain the cycle more *D-stably* than directly using the spectral radius (Nazaret et al., 2024). We prove Theorem 2 in Appendix A.

## 4.3 RECONSTRUCTION AND OPTIMIZATION

We now define score-based optimization objectives (Sun et al., 2023; Cheng et al., 2025) to train the model to identify the causal structures. We rearrange the observed time series, and build the cause series $\mathcal{X}^c = \{\mathbf{X}_t^c := [\mathbf{X}_t| \cdots |\mathbf{X}_{t-\tau}]\}_{t=\tau+1}^{T}$ and the effect series $\mathcal{Y} = \{\mathbf{Y}_t := \mathbf{X}_t\}_{t=\tau+1}^{T}$. The score-based objective claims that a causal model consistent with the data distribution should accurately reconstruct the effect series from the cause series and satisfy the acyclic constraint. Given the fine-grained causal matrices $\mathcal{W} = \{\mathbf{W}_t \in \mathbb{R}^{dm \times d\tau}\}_{t=\tau+1}^{T}$ estimated in Section 4.1, we set $m = 1$ to represent the linear causality and define the reconstructed effect series as $\hat{\mathbf{Y}}_{t:t+K} = \mathbf{W}_t \mathbf{X}_{t:t+K}^c$. Minimizing the reconstruction loss with the acyclic constraint in Section 4.2 leads to the optimization problem:

$$\underset{\mathcal{W}, \theta}{\arg\min} L = \frac{1}{NTK} \sum_{t=\tau+1}^{T-K} ||\mathbf{Y}_{t:t+K} - \hat{\mathbf{Y}}_{t:t+K}||_2^2 + \beta|\mathbf{W}_t|_1$$

$$\text{subject to } h_{norm}(\mathbf{W}_t^{ins}) = 0, t = 1, \cdots, T \qquad (7)$$

where $\theta$ is the parameter of the convolutional network and $\beta$ is a penalty coefficient. The observations in $[t, t + K]$ are reconstructed by the same matrix $\mathbf{W}_t$, which is equivalent to adding samples to learn stable $\mathbf{W}_t$ at each time step. We include the $l_1$ regularizer to promote sparse matrices and adopt the central path approach to solve this optimization.

The `DyCausal` procedure is summarized in Algorithm 1. We multiply the minimization objective by a coefficient $\mu$, and then add the acyclic constraint as a regularization to constitute a new objective. In each iteration, we reduce $\mu$ and minimize this objective. After the optimization, we prune the small elements of the causal matrices $\{\mathbf{W}_t\}_{t=\tau+1}^{T}$ with a threshold $\delta$, and obtain the dynamic causal structures $\mathcal{G}$. We discuss the identifiability of `DyCausal` in Appendix D.

**Extension to the nonlinear model.** By identifying the fully dynamic causal matrix, `DyCausal` can flexibly adapt to nonlinear causal models through parameter adjustment. Inspired by (Zheng et al., 2020b), which replaces the partial derivatives of the reconstruction model with the parameters of the first-layer network to capture nonlinear causality, we build a reconstruction model and treat the causal matrix $\mathbf{W}_t$ as its first layer. We set $m > 1$ to represent the number of hidden units and introduce additional multilayer perceptrons to reconstruct the effect series as follows:

$$\hat{\mathbf{Y}}_{t:t+K} = \mathcal{MLP}(\sigma(\mathbf{W}_t \mathbf{X}_{t:t+K}^c)) \qquad (8)$$

where $\sigma(\cdot)$ is an activation function, $\mathcal{MLP} = \{MLP_i\}_{i=1}^{d}$ is composed of $d$ independent multilayer perceptrons with input size $m$ and output size 1. We keep the optimization process in Algorithm 1, except for the additional optimization of parameters in $\mathcal{MLP}$. By reshaping the

causal matrix $\mathbf{W}_t$ from $\mathbb{R}^{dm \times d\tau}$ to $\mathbb{R}^{d \times m \times d\tau}$, we get the new weighted adjacency causal matrix $\tilde{\mathbf{W}}_t = \sqrt{\sum_{j=1}^{m}(\mathbf{W}_t[:,j,:])^2}, \tilde{\mathbf{W}}_t \in \mathbb{R}^{d \times d\tau}$.

**Extension to the ordinary differential equation model.** DyCausal can also be extended to the currently popular ordinary differential equation (ODE) (Chen et al., 2018; De Brouwer et al., 2020; Bellot et al., 2022), which is widely used in the field of biology (Zhu et al., 2012; Zheng et al., 2020a; Wang et al., 2023). The biggest difference between ODE and SEM is that ODE describes the generation process of change rates rather than changed variable values. We adjust the reconstruction process based on the nonlinear reconstruction model as follows:

$$\hat{\mathbf{Y}}_{t:t+K} = \hat{\mathbf{Y}}_{t-1:t-1+K} + \mathcal{MLP}(\sigma(\mathbf{W}_t\mathbf{X}_{t:t+K}^c)) \tag{9}$$

The reconstruction process of ODE does not involve instantaneous causality, so there is no need to penalize the cycle in the causal matrix, which is implemented in Algorithm 1 by removing $h_{norm}$ and setting the number of iterations to $r = 1$.

## 5 EXPERIMENTS

We conduct extensive experiments on synthetic and real datasets to evaluate the effectiveness of DyCausal. To compare the performance of DyCausal with all baselines, we follow existing work (Cheng et al., 2025; Zheng et al., 2018) and report three widely used metrics: TPR (true positive rate), SHD (the difference between real and estimated graphs, including the number of reversed, missing, and redundant edges), and F1 (the harmonic mean of precision and recall).

**Synthetic data.** We use the ER model (Erd6s & Rényi, 1960) to generate random ground-truth causal graphs. We then use the additive noise model and the Lorenz model (Lorenz, 1996) to simulate the structural equation and the ordinary differential equation, respectively, and generate the time series. A more detailed data generation process is given in Appendix B.1.

**Real-world data.** To demonstrate the broad applicability of DyCausal to real-world problems, we apply it to four real-world datasets: CausalTime (Cheng et al., 2024b) records time series in three real scenarios, including weather, traffic, and medical; NetSim (Smith et al., 2011) records the time series of blood-oxygen-level dependent signals in different brain regions; DREAM-3 (Prill et al., 2010) and Phoenix (Hossain et al., 2024) record the time series of gene expression levels.

**Hyperparameter settings.** We conduct experiments using the public code available for the comparison baselines. Detailed hyperparameter settings for our DyCausal and baselines are stated in Appendix B.2, which are sufficient to play the performance of the baselines. We also perform a hyperparameter analysis in Appendix C.11, which proves that we have selected the optimal hyperparameter for DyCausal.

### 5.1 RESULTS ON SYNTHETIC DATA

We first compare DyCausal against state-of-the-art methods to demonstrate its performance in dynamic temporal causal discovery, including DyCAST (Cheng et al., 2025) for partial dynamic temporal causal discovery, DYNO (short for DYNOTEARS) (Pamfil et al., 2020), NTS-NO (short for NTS-NOTEARS) (Sun et al., 2023) for static temporal causal discovery. Following Appendix B.1, we generate ground-truth causal graphs and then assign weights to each edge. To simulate dynamically changing causal structures, we let the weights of the edges in the causal graphs be the sine or cosine functions. We simulate two dynamic scenarios: one where only instantaneous causality changes over time, and the other where all causality changes over time. We generate $N = 200$ time series with length $T = 10$ and maximum lag $\tau = 1$. Figure 3 shows the evaluation of the causal structures identified by DyCausal and by each baseline at each time step.

It is clear that DyCausal outperforms other baselines and identifies the most accurate dynamic causal structures at all time steps. DYNO and NTS-NO have decreased performance at both ends of the time series. This is because they are insensitive to the changing data distribution, confuse the distribution of observations at both ends and in the middle of the series, and fail to identify dynamic causal structures. DyCAST is designed for dynamic instantaneous causality and thus robustly identifies causal structures at each time step in the corresponding scenario, but with less accuracy than

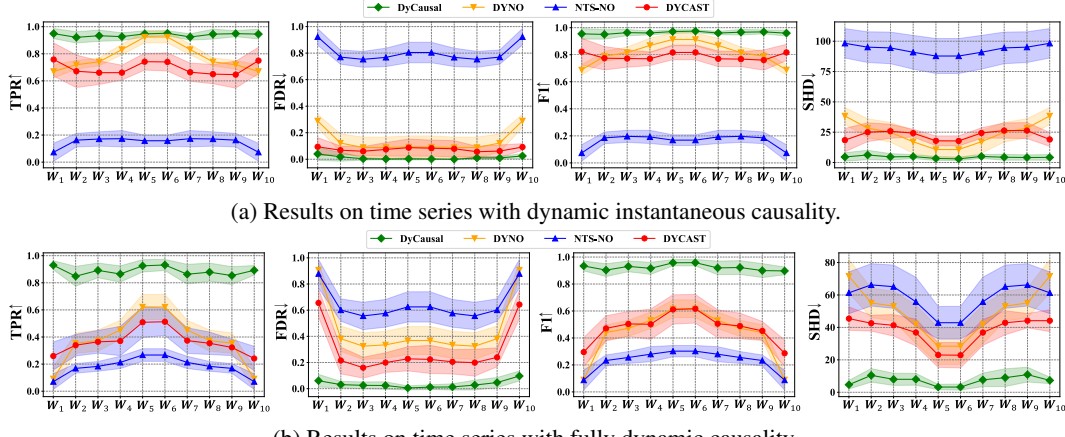

(a) Results on time series with dynamic instantaneous causality.

(b) Results on time series with fully dynamic causality.

Figure 3: Results on time series with dynamic causality. $\{\mathbf{W}_1, \cdots, \mathbf{W}_{10}\}$ represent the causal graphs estimated at each time step.

Table 1: Results on time series with dynamic causality. $\mathbf{W}_1$ and $\mathbf{W}_{50}$ represent the causal graphs estimated at both ends of the time series. $\mathbf{W}_{25}$ represents the causal graph estimated in the middle of the time series. We omit the percentage signs % of TPR and F1 to simplify the representation.

| $t$ | Methods | 20 nodes | | | 80 nodes | | |
|---|---|---|---|---|---|---|---|
| | | TPR↑ | F1↑ | SHD↓ | TPR↑ | F1↑ | SHD↓ |
| $\mathbf{W}_1$ | DyCausal | **89.54±6.98** | **91.43±5.60** | **7.40±5.39** | **89.10±1.70** | **93.85±1.31** | **18.20±3.87** |
| | DYCAST | 25.98±10.33 | 29.51±10.67 | 45.43±7.21 | - | - | - |
| | DYNO | 3.59±3.18 | 3.52±2.90 | 78.40±10.75 | 0.35±0.70 | 0.34±0.68 | 294.90±28.51 |
| | NTS-NO | 3.89±1.57 | 4.03±1.69 | 72.33±12.04 | 0.77±1.01 | 0.78±1.02 | 294.20±15.55 |
| $\mathbf{W}_{25}$ | DyCausal | **90.93±6.01** | **94.38±3.90** | **4.10±2.88** | **91.01±1.97** | **95.20±1.18** | **13.40±2.15** |
| | DYCAST | 50.88±11.29 | 61.12±10.91 | 23.00±7.63 | - | - | - |
| | DYNO | 71.64±6.77 | 69.20±6.96 | 25.00±6.91 | 69.49±4.70 | 72.81±4.90 | 82.90±17.22 |
| | NTS-NO | 30.99±5.24 | 33.00±7.37 | 51.56±10.74 | 32.36±4.23 | 33.28±4.07 | 196.10±16.31 |
| $\mathbf{W}_{50}$ | DyCausal | **87.29±6.60** | **90.38±5.99** | **8.20±6.06** | **87.55±2.89** | **92.52±1.78** | **22.00±4.60** |
| | DYCAST | 24.15±8.97 | 28.64±9.87 | 44.14±6.75 | - | - | - |
| | DYNO | 3.59±3.18 | 3.52±2.90 | 78.40±10.75 | 0.35±0.70 | 0.34±0.68 | 294.90±28.51 |
| | NTS-NO | 3.89±1.57 | 4.03±1.69 | 72.33±12.04 | 0.77±1.01 | 0.78±1.02 | 294.20±15.55 |

`DyCausal`. Limited by the ordinary differential equation to model the time-varying trajectory of causality, DyCAST struggles to identify causal structures with drastically changing weights. Only `DyCausal` succeeds in time series with fully dynamic causality, thanks to coarse-to-fine modeling of time-varying trajectory with instantaneous and lagged causality.

**Performance on time series with dynamic causal structures.** We run `DyCausal` and baselines on longer time series with larger causal graphs to further demonstrate the superior performance of `DyCausal`. Specifically, we set the length of the time series as $T = 50$ and the number of nodes in the causal graphs at $d = \{20, 80\}$. Table 1 presents the evaluation of the causal graphs identified by `DyCausal` and the baselines at three key time steps $t = \{1, 25, 50\}$. We can see that `DyCausal` is significantly superior to all baselines and its advantage is especially prominent at both ends of the series ($t = 1$ and $t = 50$). Compared to the static methods DYNO and NTS-NO, DYCAST identifies partially dynamic causal graphs but fails to learn causal matrices with opposite weights at both ends of the series. This indicates that the ordinary differential equation modeling the time-varying trajectory of the causal matrix cannot identify the weights with large variations. In addition, DYCAST models the causal graph at each time step and thus fails to converge on graphs with 80 nodes. In contrast, `DyCausal` only encodes causal graphs for partial time steps and then refines them to each time step by linear interpolation, improving stability and accuracy. Appendix C.13 elaborately presents the evaluation of the instantaneous and lagged causal graphs identified by `DyCausal` at each time step, further demonstrating the superior performance of `DyCausal` in identifying fully dynamic causal relationships.

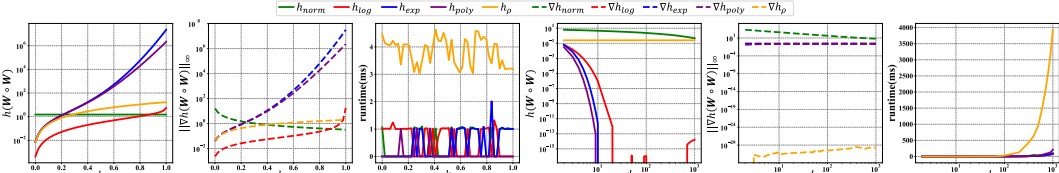

Figure 4: Comparison of acyclic constraints and their gradients, and runtime.

We further provide ablation experiments in Appendix C.2 to prove that linear interpolation in `DyCausal` is necessary for identifying dynamic causality. We also evaluate `DyCausal` on time series with non-smooth dynamic causality. The results in Appendix C.12 indicate that `DyCausal` can still effectively identify causal graphs with considerable accuracy. We run `DyCausal` on datasets with more variables, fewer samples, longer lag and different series length and noise strength in Appendices C.3~C.8, and the results show that `DyCausal` is scalable and robust.

**Performance on time series with static causal structures.** We conduct comparative experiments to show that `DyCausal` is also capable of identifying static causal structures from time series. In particular, we synthesize time series with three models: the linear and nonlinear structure equation models (SEM) and the ordinary differential equation (ODE). Comparison baselines include SVAM (Hyvärinen et al., 2008), PCMCI (Runge et al., 2019), and TECDI (Li et al., 2023) for all models, DYNO for linear SEM, NTS-NO for nonlinear SEM and NGM (Bellot et al., 2022) for ODE. We generate ground-truth causal graphs and time series with length $T = 1000$ and maximum lag $\tau = 2$ as described in Appendix B.1. To adapt to algorithms that require multiple time series, we divide the original series into $N = 20$ time series with length $T = 50$, which ensures consistent sample sizes and fair comparisons. We report the results of `DyCausal` and all baselines in Appendix C.9. The results show that `DyCausal` consistently improves accuracy under three models and attains a significant speed-up. We also compare the acyclic constraint $h_{norm}$ against with the original constraint $h_{log}$ in Appendix C.10, and prove that the improvement of $h_{norm}$ is effective and necessary.

## 5.2 EVALUATION OF THE ACYCLIC CONSTRAINT

We have proved in Theorem 2 and Appendix A that $h_{norm}$ meets the stability criteria. We provide empirical results to further demonstrate the superiority of $h_{norm}$ over existing acyclic constraints, including $h_{exp}$ (Zheng et al., 2018), $h_{poly}$ (Yu et al., 2019), $h_{log}$ (Bello et al., 2022), and $h_\rho$ (Nazaret et al., 2024)). We compare these acyclic constraints in two types of matrices: uniform random matrices in $[0, k]$, and cycle graphs with $d$ nodes where each edge weight is $0.5$ or $-0.5$. Figure 4 shows the curves of the acyclic constraints and their gradients, and runtime varying with $k$ and $d$.

As $k$ and $d$ increases, $h_{exp}$ and $h_{poly}$ rapidly grow to infinity or vanish to 0, reflecting that they are *E-unstable* and *V-unstable*. $h_{log}$ performs robustly on uniform random matrices but its performance decreases rapidly with increasing $d$ on cyclic graphs. This is because small weights reduce the sensitivity of $h_{log}$ to long cycles. In contrast, $h_{norm}$ is robust against the rapid expansion and attenuation, and remains stable and efficient in all experimental settings. Although $h_\rho$ is similarly stable on both types of matrices, it has a very small gradient and exponentially increasing runtime on cyclic graphs. We observe that cycle graphs have close eigenvalues, which could be the reason for the disappearance of $\nabla h_\rho$. A small $\nabla h_\rho$ means that $h_\rho$ is prone to getting stuck in local optimal solutions and cannot effectively punish the cycle. Therefore, the acyclic constraint $h_{norm}$ is the most stable and efficient, and also the most effective in punishing cycles.

## 5.3 RESULTS ON A REAL CASE

We evaluate `DyCausal` and other baselines on CausalTime (Cheng et al., 2024b). In particular, we introduce an additional baseline, CUTS+ (Cheng et al., 2024a), which targets at high-dimensional and sparse real-world data, and JRNGC (Zhou et al., 2024), which identify causal structures with the Jacobian matrix. Table 2 shows a detailed comparison of the results.

We observed that `DyCausal` achieves the best Precision and F1. Higher precision means that `DyCausal` can identify more reliable and trustworthy causal structures (with most of the edges cor-

Table 2: Results on CausalTime. The boldfaced and underlined results respectively highlight the methods that perform the first and second best among all compared methods. We omit the percentage signs % of Precision and F1 to simplify the representation.

| Methods | Traffic | | AQI | | Medical | |
|---|---|---|---|---|---|---|
| | Precision↑ | F1↑ | Precision↑ | F1↑ | Precision↑ | F1↑ |
| DyCausal | **87.41±3.85** | **54.81±2.51** | **75.31±1.43** | **63.16±1.23** | **83.21±2.75** | **56.86±1.35** |
| CUTS+ | 80.10±3.16 | 45.21±1.74 | 65.60±3.47 | 58.20±2.44 | 44.82±3.54 | 30.23±1.85 |
| JRNGC | 68.04±5.56 | 52.04±2.41 | 72.34±3.43 | 55.36±1.61 | 55.56±0.63 | 52.05±0.75 |
| NTS-NO | 67.43±5.35 | 46.07±1.85 | 72.70±2.99 | 40.09±1.29 | 44.80±2.13 | 34.27±1.89 |
| DYNO | 47.63±3.26 | 45.34±2.92 | 62.55±1.34 | 44.60±0.75 | 48.40±2.07 | 31.70±1.25 |
| SVAM | 45.45±1.39 | 48.26±1.14 | 55.22±0.81 | 50.26±0.78 | 49.33±0.41 | 41.63±0.24 |
| PCMCI | 22.61±2.98 | 25.30±3.28 | 31.74±4.51 | 39.26±7.72 | 32.71±5.21 | 37.39±5.46 |
| NGM | 21.21±2.15 | 30.76±3.17 | 28.38±1.64 | 31.76±2.16 | 37.54±1.28 | 44.15±1.70 |
| TECDI | 70.05±6.00 | 44.01±1.66 | 56.18±2.86 | 47.44±2.35 | 44.22±3.58 | 36.51±2.99 |

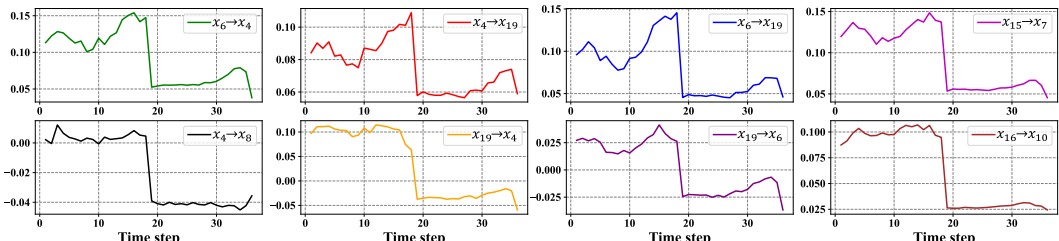

Figure 5: Estimated dynamic causal structures on traffic subset. We observed that: (i) causality $x_6 \rightarrow x_4$, $x_4 \rightarrow x_{19}$, $x_6 \rightarrow x_{19}$, $x_{15} \rightarrow x_7$ and $x_{19} \rightarrow x_6$ are dynamic up to the time $t = 20$ and static after;(ii) causality $x_4 \rightarrow x_8$, $x_{19} \rightarrow x_4$ and $x_{16} \rightarrow x_{10}$ are static before and after $t = 20$, but with different causal strengths. In particular, the strength of $x_4 \rightarrow x_8$ is almost zero before $t = 20$, which means that $x_4 \rightarrow x_8$ can only be identified from time series after $t = 20$.

rect) when facing datasets without verifiable causal graphs. CUTS+ identifies the summary causality without limiting the window length and is inferior to DyCausal. This indicates that the window causality is closer to the true causality of CausalTime than summary causality. Figure 5 visualizes the dynamic causal structure identified by DyCausal in the traffic subset, which records the flow changes of 20 traffic nodes within a day. It can be seen that the dynamic causality is clearly divided into two change patterns by time step $t = 20$. DyCausal can accurately capture the dynamic changes in causal relationships that other algorithms often treat as noise, preserving true structures while avoiding the inclusion of redundant ones. Appendix C.14 provides the results of DyCausal in other real datasets, NetSim (Smith et al., 2011), DREAM-3 (Prill et al., 2010), and Phoenix (Hossain et al., 2024), where DyCausal identifies the most accurate causal graphs.

## 6 CONCLUSION

In this paper, we propose DyCausal, a novel framework to overcome inefficiency and instability in fully dynamic temporal causal discovery. Compared to previous methods, DyCausal learns the time-varying trajectory of causality from coarse to fine with the most advanced performance. The theoretically and empirically stable acyclicity constraint, enforced via the matrix norm, further enhances both the robustness and efficiency of DyCausal. Extensive experiments demonstrate that DyCausal consistently performs well in dynamic, partially dynamic, and static time series, adapts well to real-world datasets, and is sensitive to the fluctuation and evolution of causality.

This work assumes that the time series is regular. Although previous work (Gong et al., 2015; Iseki et al., 2019) attempted to solve irregular time series, this is not the focus of this work. DyCausal can use the existing completion strategies (Cao et al., 2018; Cini et al., 2022; Jin et al., 2024) to impute irregular time series and then identify dynamic causal structures. In the future, we plan to expand DyCasual to a causal system where variables dynamically participate or leave.

## 7 ACKNOWLEDGMENTS

This work is partially supported by National Key Research and Development Program of China (No. 2024YFF1206604), National Natural Science Foundation of China (62531013, 62272276 and 62432006), Shandong Provincial Natural Science Foundation (No. ZR2024JQ001 and ZR2025LZH004) and Taishan Scholars Program (No. tsqn202408317).

## 8 ETHICS STATEMENT

We promise that we have read the ICLR Code of Ethics, and this article has not raised any questions regarding the Code of Ethics.

## 9 REPRODUCIBILITY STATEMENT

We promise that `DyCausal` is reproducibility. We provide the code of `DyCausal` at https://www.sdu-idea.cn/codes.php?name=DyCausal, including the code to run, synthesize datasets, and perform experiments. The code also includes the real datasets used in the experiments.

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

## A  PROOF OF STABILITY FOR THE ACYCLIC CONSTRAINT

We recall Theorem 2 and prove it.

**Theorem 2. *Stability*.** *The acyclic constraint $h_{norm}$ is stable. For almost every $\mathbf{A} \in \mathbb{R}_{\geq 0}^{d \times d}$:*

   (i) ***E-stable*** $h_{norm}(k\mathbf{A}) = \mathcal{O}_{k \to \infty}(1)$.

   (ii) ***V-stable*** $h_{norm}(\mathbf{A}) \neq 0 \Rightarrow h_{norm}(k\mathbf{A}) = \Omega_{k \to 0^+}(1)$.

   (iii) ***D-stable*** $h_{norm}$ and $\nabla h_{norm}$ are defined almost everywhere.

**Proof.** Since the 1-norm is the maximum sum of the absolute values of the elements in each column of the matrix, if the matrix $\mathbf{A}$ is multiplied by the constant $k$, its 1-norm is also multiplied by the constant $k$. Therefore, we have the following conclusions:

   • **E-stable.** For any $k > 0$ and matrix $\mathbf{W}$, $h_{norm}(k\mathbf{A}) = -\log \det(\alpha I - \frac{k\mathbf{A}}{||k\mathbf{A}||_1}) + d \log \alpha = -\log \det(\alpha I - \frac{\mathbf{A}}{||\mathbf{A}||_1}) + d \log \alpha = h_{norm}(\mathbf{A}) = \mathcal{O}_{k \to +\infty}(1)$.

   • **V-stable.** For any $k > 0$ and matrix $\mathbf{A}$ such that $h_{norm}(\mathbf{A}) > 0$, $h_{norm}(k\mathbf{A}) = -\log \det(\alpha I - \frac{k\mathbf{A}}{||k\mathbf{A}||_1}) + d \log \alpha = -\log \det(\alpha I - \frac{\mathbf{A}}{||\mathbf{A}||_1}) + d \log \alpha = h_{norm}(\mathbf{A}) = \Omega_{k \to 0^+}(1)$.

According to Theorem 1, $h_{norm}$ inherits the excellent characteristics of $h_{log}$ and $\nabla h_{norm}$ is well defined within the optimizable space. Furthermore, the spectral radius of the matrix scaled by the 1-norm is always within the optimizable space ($\rho(\frac{\mathbf{A}}{||\mathbf{A}||_1}) < 1^+$). Therefore, $h_{norm}$ and its gradient $\nabla h_{norm}$ are well defined everywhere, and $h_{norm}$ is **D-stable**.

## B EXPERIMENTAL DETAILS

### B.1 DATA GENERATION

In this section, we provide a detailed process for generating the simulation data in experiments.

**Ground-truth graph.** Given the number of variables $d$, the maximum lagged length $\tau$ and edge density $e$, we randomly generate ground-truth graphs. Specifically, we use the ER model (Erd6s & Rényi, 1960) to generate a DAG with $d$ nodes and $e \times d$ edges as the instantaneous causal graph $\mathbf{G}^{ins}$. We then set $\tau$ zero matrices in $\mathbb{R}^{d \times d}$. In each zero matrix, the elements are set to 1 with probability $e/d$ and the lagged causal graph $\mathbf{G}^{lag}$ is obtained. We vertically concatenate the instantaneous and lagged causal graphs to obtain the complete temporal causal graph $\mathbf{G}$. In our experiments, we always set the edge density $e = 2$.

**Linear structural equation model.** We use the linear additive noise model to simulate the linear structural equation to generate data as follows:

$$\mathbf{X}_t = \mathbf{W}[\mathbf{X}_t | \mathbf{X}_{t-1} | \cdots | \mathbf{X}_{t-\tau}] + \epsilon_t \tag{A1}$$

where $\epsilon_t$ is sampled from the Gaussian distribution $N(0, 1)$. To assign weights to each nonzero element in the ground-truth causal graph, we uniformly sample weights from the interval $[-0.5, -0.3] \cup [0.3, 0.5]$ to obtain a causal weight matrix $\mathbf{W}$. Furthermore, we multiply each weight from time $t$ to $t - \tau$ by a decay parameter $(1/\eta)^p$, where $\eta > 1$ and $p \in [0, \tau]$. This strategy is widely used by existing methods (Pamfil et al., 2020; Cheng et al., 2025) to reduce the influence of variables that are further back in time relative to current time. In our experiments, we always set $\eta = 1.5$.

**Nonlinear structural equation model.** We use the nonlinear additive model parameterized by multilayer perceptrons to simulate the nonlinear structural equation to generate data as follows:

$$x_{t,i} = MLP(\sigma(MLP(PA(x_{t,i})))) + \epsilon_t \tag{A2}$$

where $\epsilon$ is sampled from the Gaussian distribution $N(0, 1)$, $\sigma(\cdot)$ is a sigmoid function and $PA(x_{t,i})$ are parents of $x_{t,i}$ in the interval $[t - \tau, t]$. We set multilayer perceptrons with 1 hidden layer and 100 hidden units and uniformly sample perceptron parameters from the interval $[-2, -0.5] \cup [0.5, 2]$.

**Ordinary differential equation model.** We use the Lorenz model (Lorenz, 1996), a widely used ordinary differential model, to generate data as follows:

$$dx_{t,i}/dt = -x_{t,i-1}(x_{t,i-2} - x_{t,i+1}) - x_{t,i} + F + dw \tag{A3}$$

where $F$ is a constant and $dw$ is sampled from the Gaussian distribution $N(0, 0.05)$. We randomly sample data from the Gaussian distribution $N(0, 0.01)$ as observed data at time $t = 0$, and then gradually generate data at time $t > 0$ according to the above model $x_{t+1,i} = x_{t,i} + dx_{t,i}$. In our experiments, we always set $F = 5$.

**Dynamic model.** Given the causal weight matrix $\mathbf{W}_0$, we set its weights to be a time-indexed sine or cosine function to simulate the dynamic causality. Specifically, we uniformly sample a probability matrix $\mathbf{W}_p \in \mathbb{R}^{d\tau \times d}$ and define the dynamic matrix $\mathbf{W}_t$ as follows:

$$\mathbf{W}_t[i, j] = \begin{cases} \mathbf{W}_0[i, j]\cos((\pi/T) \cdot t) & \text{if } \mathbf{W}_p[i, j] < 0.5 \\ \mathbf{W}_0[i, j]\sin((\pi/T) \cdot t) & \text{if } \mathbf{W}_p[i, j] > 0.5 \end{cases} \tag{A4}$$

where $T$ is the length of the series. Given the above weight matrix, the observed data at time $t$ is only generated by $\mathbf{W}_t$.

### B.2 PARAMETER SETTING

In this section, we provide the detailed parameter settings of all algorithms used in experiments. For fair comparison, we search for parameters such as regularization coefficient, learning rate, and pruning threshold that do not affect the model size to achieve the best performance of the baselines. In particular, we set `DyCausal`, NGM and NTS-NOTEARTS to have the same reconstruction model size to fairly compare their performance. All experiments were run on the same server (Ubuntu 18.04.5, Intel Xeon Gold 6248R and Nvidia RTX 3090). The code of `DyCausal` is shared at https://www.sdu-idea.cn/codes.php?name=DyCausal.

- DyCausal
    - The window size $K = 10$ for static and $K = 2$ for dynamic
    - The sliding step size $S = 5$ for static and $S = 4$ for dynamic
    - The model contains 1 hidden layer and 10 hidden units for nonlinear data
    - The learning rate $lr = 0.005$
    - The sparse term coefficient $\beta = 0.05$
    - The pruning threshold $\delta = 0.3$
- PCMCI
    - The conditional independence test uses the Pearson correlation coefficient
    - Significance P value $PC_\alpha = 0.05$
- SVAM
    - The pruning threshold $\delta = 0.3$
- NGM
    - The Lasso regularization coefficient $\beta = 0.05$
    - The model contains 1 hidden layer and 10 hidden units
    - The batch size $batch\_size = 20$
    - The predicted length $horizon = 5$
    - The pruning threshold $\delta = 0.1$
- DYNOTEARS
    - The sparse term coefficient $\beta = 0.1$
    - The pruning threshold $\delta = 0.1$
- NTS-NOTEARS
    - The sparse term coefficient $\beta = 0.01$
    - The model contains 1 hidden layer and 10 hidden units
    - The pruning threshold $\delta = 0.5$
- TECDI
    - The model contains 2 hidden layer, and each layer contains 16 hidden units
    - The learning rate $lr = 0.005$
    - The Regularization coefficient $\beta = 0.3$
- DYCAST
    - The model contains 2 hidden layer, and each layer contains 16 hidden units
    - The learning rate $lr = 0.001$
    - The hidden units is $0.8d$
    - The pruning threshold $\delta = 0.1$

## C  ADDITIONAL EXPERIMENTS

### C.1  RESULTS ON TIME SERIES WITH DYNAMIC CAUSAL STRUCTURES

We demonstrate the performance of DyCausal on time series with dynamic causality. We follow Appendix B.1 to generate ground-truth causal graphs and then assign time-varying weights to each edge. We let the weights of the edges in the causal graphs be the sine or cosine functions of the time index, thereby simulating the dynamic causal relationships. We generate $N = 200$ time series with length $T = 50$ and maximum lag $\tau = 1$. Table A1 shows the results of DyCausal and the baselines at three key time points $t = \{1, 25, 50\}$.

We can see that DyCausal is significantly superior to all baselines and its advantage is especially prominent at both ends of the series ($t = 1$ and $t = 50$). Compared to the static methods DYNO and NTS-NO, DYCAST identifies partially dynamic causal graphs but fails to learn causal matrices with opposite weights at both ends of the series. This indicates that the ordinary differential equation

Table A1: Results on time series with dynamic causality. $\mathbf{W}_1$ and $\mathbf{W}_{50}$ represent the causal graphs estimated at both ends of the time series. $\mathbf{W}_{25}$ represents the causal graph estimated in the middle of the time series. We omit the percentage signs % of TPR and F1 to simplify the representation.

| $t$ | Methods | 20 nodes | | | 80 nodes | | |
|---|---|---|---|---|---|---|---|
| | | TPR↑ | F1↑ | SHD↓ | TPR↑ | F1↑ | SHD↓ |
| $\mathbf{W}_1$ | DyCausal | **89.54±6.98** | **91.43±5.60** | **7.40±5.39** | **89.10±1.70** | **93.85±1.31** | **18.20±3.87** |
| | DYCAST | 25.98±10.33 | 29.51±10.67 | 45.43±7.21 | - | - | - |
| | DYNO | 3.59±3.18 | 3.52±2.90 | 78.40±10.75 | 0.35±0.70 | 0.34±0.68 | 294.90±28.51 |
| | NTS-NO | 3.89±1.57 | 4.03±1.69 | 72.33±12.04 | 0.77±1.01 | 0.78±1.02 | 294.20±15.55 |
| $\mathbf{W}_{25}$ | DyCausal | **90.93±6.01** | **94.38±3.90** | **4.10±2.88** | **91.01±1.97** | **95.20±1.18** | **13.40±2.15** |
| | DYCAST | 50.88±11.29 | 61.12±10.91 | 23.00±7.63 | - | - | - |
| | DYNO | 71.64±6.77 | 69.20±6.96 | 25.00±6.91 | 69.49±4.70 | 72.81±4.90 | 82.90±17.22 |
| | NTS-NO | 30.99±5.24 | 33.00±7.37 | 51.56±10.74 | 32.36±4.23 | 33.28±4.07 | 196.10±16.31 |
| $\mathbf{W}_{50}$ | DyCausal | **87.29±6.60** | **90.38±5.99** | **8.20±6.06** | **87.55±2.89** | **92.52±1.78** | **22.00±4.60** |
| | DYCAST | 24.15±8.97 | 28.64±9.87 | 44.14±6.75 | - | - | - |
| | DYNO | 3.59±3.18 | 3.52±2.90 | 78.40±10.75 | 0.35±0.70 | 0.34±0.68 | 294.90±28.51 |
| | NTS-NO | 3.89±1.57 | 4.03±1.69 | 72.33±12.04 | 0.77±1.01 | 0.78±1.02 | 294.20±15.55 |

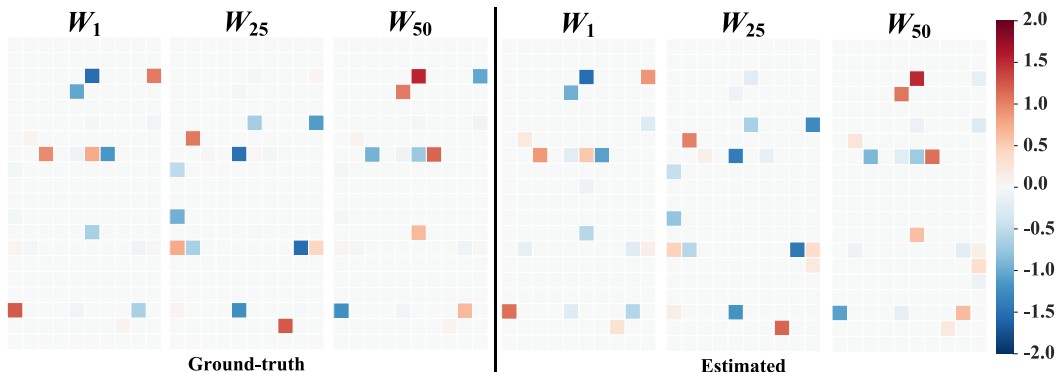

Figure A1: Results on dynamic settings with $d = 10$ variables and $\tau = 1$ maximum lag. $\mathbf{W}_1$ and $\mathbf{W}_{50}$ represent the causal graphs estimated at both ends of the time series. $\mathbf{W}_{25}$ represents the causal graph estimated in the middle of the time series. The left and right parts represent the ground-truth and the estimated causal matrices. Red and blue represent the positive and negative matrix weights respectively, and the darker the color, the greater the absolute value of the weight. The estimated matrices are basically consistent with the ground-truth matrices.

modeling the time-varying trajectory of the causal matrix cannot identify the weights with large variations. In addition, DYCAST models the causal graph at each time step and thus fails to converge on graphs with 80 nodes. In contrast, DyCausal only encodes causal graphs for partial time steps and then refines them to each time step by linear interpolation, improving stability and accuracy. We further compare the causal matrices estimated by DyCausal and the ground-truth causal matrices in Figure A1, where DyCausal not only identifies the causal graphs, but also restores the weights of the causal matrices.

## C.2 ABLATION FOR LINEAR INTERPOLATION IN DYNAMIC SETTINGS

Linear interpolation is necessary for large-scale causal graphs and long time series. It enables the coarse-grained causal matrices learned by the window to jointly participate in the training; otherwise, each matrix can only be trained in isolation based on the data within the corresponding window. We introduce an ablation baseline DyCausal w/o inter in dynamic settings, which only allows the causal matrices encoded by the sliding convolutional window to participate in the series reconstruction without approximating the fine-grained time-varying trajectory of the causal matrix through linear interpolation. Table A2 shows the performance comparison between DyCasual and DyCausal w/o inter.

Table A2: Results on time series with dynamic causality. $\mathbf{W}_1$ and $\mathbf{W}_{50}$ represent the causal graphs estimated at both ends of the time series. $\mathbf{W}_{25}$ represents the causal graph estimated in the middle of the time series. We omit the percentage signs % of TPR and F1 to simplify the representation.

| $t$ | Methods | 10 nodes | | | 20 nodes | | |
|---|---|---|---|---|---|---|---|
| | | TPR↑ | F1↑ | SHD↓ | TPR↑ | F1↑ | SHD↓ |
| $\mathbf{W}_1$ | DyCausal | **90.35±6.62** | **91.42±4.39** | **3.20±1.40** | 89.86±3.83 | **92.26±2.98** | **6.00±2.37** |
| | w/o inter | 75.67±22.20 | 72.98±23.78 | 12.50±13.86 | **90.06±5.45** | 92.18±4.12 | 6.10±3.67 |
| $\mathbf{W}_{25}$ | DyCausal | **90.41±7.88** | **92.65±7.73** | **3.10±3.24** | **91.09±5.06** | **93.64±5.47** | **4.90±4.57** |
| | w/o inter | 81.66±15.05 | 83.25±20.30 | 7.50±11.15 | 88.69±10.05 | 91.28±9.88 | 6.40±7.30 |
| $\mathbf{W}_{50}$ | DyCausal | **89.62±6.94** | **90.81±3.67** | **3.50±1.43** | **87.75±6.53** | **89.40±4.84** | 8.30±4.03 |
| | w/o inter | 76.26±16.17 | 72.83±20.28 | 12.80±13.31 | 87.45±12.35 | 87.78±11.21 | 9.50±8.48 |

Table A3: Results for settings $K = 4, S = 1$ and $K = 1, S = 1$.

| | DyCausal | | | $K = 4, S = 1$ | | | $K = 1, S = 1$ | | |
|---|---|---|---|---|---|---|---|---|---|
| | TPR | F1 | SHD | TPR | F1 | SHD | TPR | F1 | SHD |
| $\mathbf{W}_1$ | 91.56 | 92.73 | 63.0 | 90.54 | 90.95 | 72.0 | 83.33 | 20.48 | 2485.0 |
| $\mathbf{W}_{25}$ | 96.70 | 98.32 | 13.0 | 95.47 | 97.68 | 18.0 | 89.22 | 30.22 | 1644.0 |
| $\mathbf{W}_{50}$ | 94.18 | 94.18 | 46.0 | 90.09 | 92.59 | 64.0 | 85.68 | 20.76 | 2512.0 |
| runtime (s) | 2926.0 | | | 3527.0 | | | 1970.0 | | |

The results show that DyCausal w/o inter achieves a worse performance than DyCasual. This shows that linear interpolation in DyCausal is not only a core component for approximating fine-grained dynamic causality, but also plays an important role in identifying dynamic causal discovery. Linear interpolation links the isolated causal matrices inferred in each sliding step, enabling data that are originally not within the sliding window to participate in the reconstruction process and guide the learning of the dynamic causal matrix. Therefore, DyCausal achieves improved precision and stability in dynamic causal discovery. On $N = 200$ time series with length $T = 50$ and $d = 200$ nodes, we further estimate the causal graph at each time step under two settings. One sets the window size $K = 4$ and the sliding step $S = 1$. The other sets both the window size and the sliding step to one. The results are reported in Table A3:

The results show that DyCausal runs for a longer time in the former setting. That is because the model encodes more matrices and calculates the acyclic constraint term more times. DyCausal achieves reduced results in the latter setting. This is because the sample size within a window of size 1 is too small to support the model in learning causal graphs with 200 nodes.

## C.3 SCALABILITY TO CAUSAL GRAPHS WITH MORE NODES

We run DyCausal on ground-truth causal graphs with more nodes to demonstrate its scalability for large-scale data. We continue to use the dynamic settings in Section C.1 and expand the number of nodes to $d = \{50, 100, 200, 300\}$. Figure A2 shows the results of DyCausal at three key time points $t = \{1, 25, 50\}$.

We can see that DyCausal maintains excellent performance on more nodes with TPR greater than $0.78$ and F1 greater than $0.82$, which proves the scalability of DyCausal in handling large-scale causal graphs. It is important to note that with increasing number of nodes, we reduce the learning rate to accurately learn causal structures. The combinations of the number of nodes and the learning rate are $(50, 5e - 5), (100, 1e - 5), (200, 3e - 6), (300, 1.5e - 6)$. This may be because a smaller learning rate can promote the convergence of the model to identify accurate causal structures.

## C.4 ROBUSTNESS TO FEWER SAMPLES

We run DyCausal on datasets with fewer samples to demonstrate the robustness of DyCausal to sample size. We also follow the dynamic settings in Section C.1 and adjust the sample size $N = \{20, 50, 100, 150\}$. Figure A3 shows the results of DyCausal on graphs with $d = 50$ nodes at three key time points $t = \{1, 25, 50\}$.

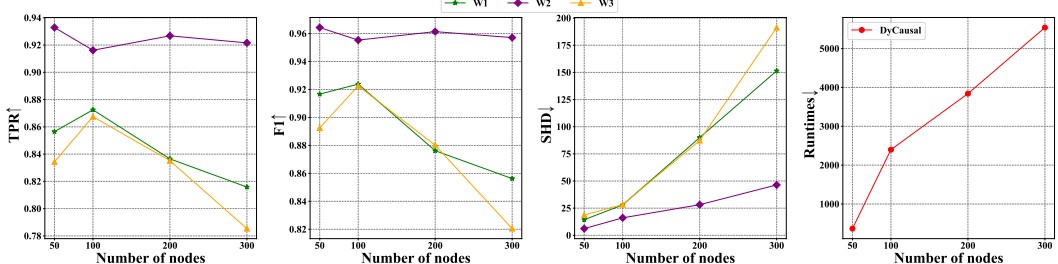

Figure A2: Results of DyCausal on datasets with number of nodes $d = \{50, 100, 200, 300\}$.

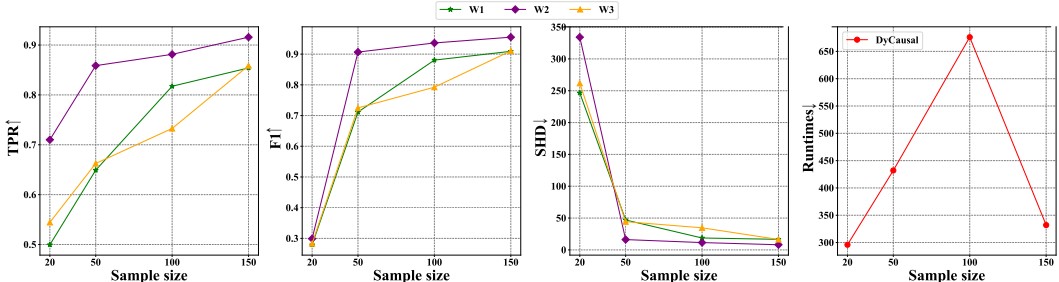

Figure A3: Results of DyCausal on datasets with fewer samples $N = \{20, 50, 100, 150\}$.

We can see that DyCausal achieves a TPR of greater than 0.5 with only 20 samples and maintains a similar performance with 50 samples as it does with 100 and 150 samples. Surprisingly, the runtime of DyCausal for 150 samples is less than that for 50 and 100 samples. This may be related to the memory scheduling of the GPU. Despite this, DyCausal still provides an acceptable runtime (less than 11 minutes). All of the above indicate that DyCausal has excellent robustness with fewer samples.

### C.5 ROBUSTNESS TO DIFFERENT TIMES SERIES LENGTHS

Still under the dynamic settings in Section C.1, we change the time series length to verify the robustness of DyCausal to different series lengths. Figure A4 shows the results of DyCasual on datasets with time series lengths $T = \{10, 30, 50, 70, 90\}$.

As the length of the time series increases, DyCausal achieves gradually improved and stable performance. In shorter time series, the data distribution changes more drastically within the sliding window, resulting in decreased performance. However, DyCausal still achieves TPR and F1 around 0.7 with $T = 10$. The runtime of DyCausal at $T = 70, 90$ is significantly longer than at $T = 10, 30, 50$. One possible reason is that longer time series occupy more memory, leading to more memory swap operations.

### C.6 ROBUSTNESS TO DIFFERENT NOISE STRENGTHS

We further verify the influence of noise strength on DyCausal. In the dynamic settings in Section C.1, we set the noise strength to $\{0.5, 1.0, 2.5, 5.0\}$, and report the results of DyCausal under these settings in Figure A5.

Larger noise, except for slightly increasing the runtime of DyCausal, does not significantly affect the performance of DyCausal. In contrast, DyCausal achieves a decrease in TPR and F1 with noise strength 0.5. This may be because smaller noise makes it difficult for the model to distinguish between direct and indirect causal effects, leading to the identification of inaccurate causal structures. The above results prove that DyCausal perform stably under different noise strengths.

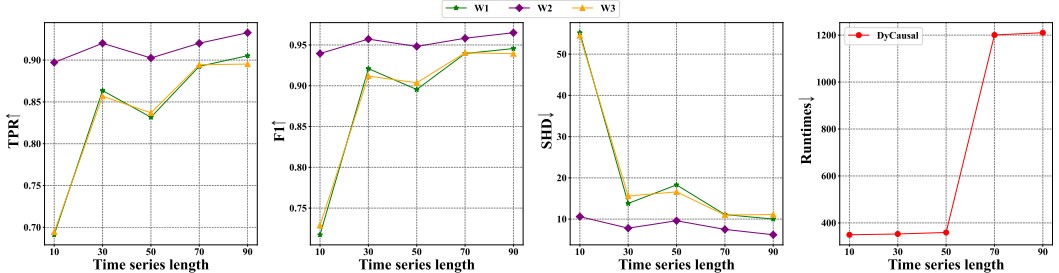

Figure A4: Results of `DyCausal` on datasets with times series lengths $T = \{10, 30, 50, 70, 90\}$.

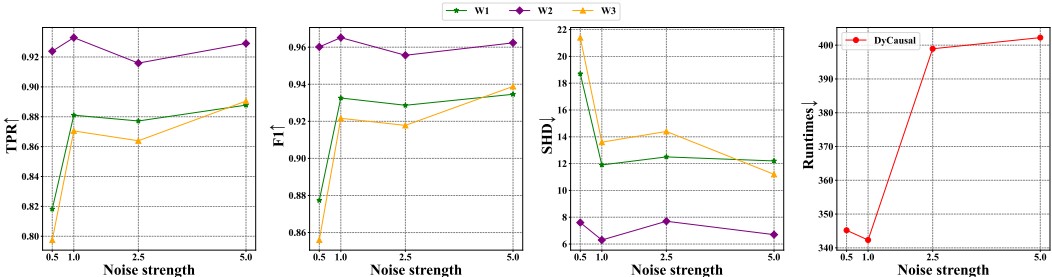

Figure A5: Results of `DyCausal` on datasets with noise strength $\{0.5, 1.0, 2.5, 5.0\}$.

### C.7 ROBUSTNESS TO LONGER LAGS

We run `DyCausal` on datasets with longer maximum lags to verify its robustness against different lags. In the dynamic settings in Section C.1, we set the number of variables $d = 50$ and the maximum lags $\tau = \{1, 2, 3, 4, 5\}$, and report the results of `DyCausal` in Figure A6.

We can observe that `DyCausal` maintains a stable performance under longer lag. When the maximum lag $\tau = 5$, `DyCausal` achieves a true positive rate of around 0.7 and an F1 of around 0.75.

### C.8 ROBUSTNESS TO UNKNOWN LAGS

We further explore the robustness of `DyCausal` to unknown lags. In the same dynamic settings, we set the maximum lags $\tau = 3$ to generate time series. We then let `DyCausal` learn temporal causal structures with estimated maximum lags $\tau = \{1, 2, 3, 4, 5\}$. For the case with $\tau < 3$, we fill the learned causal graphs with 0, and for the case with $\tau > 3$, we only retain the structures of the graphs where the lag is less than or equal to 3. Figure A7 shows the results of `DyCausal`.

We can clearly see that as the estimated lag increases, `DyCausal` achieves a higher true positive rate. Although an overly larger estimated lag also reduces performance, we suggest choosing a slightly larger estimated lag for `DyCausal` when the maximum lag is unknown.

### C.9 RESULTS ON TIME SERIES WITH STATIC CAUSAL STRUCTURES

We generate ground-truth causal graphs and time series with length $T = 1000$ and maximum lag $\tau = 2$ as described in Appendix B.1. To adapt to algorithms that require multiple time series, we divide the original series into $N = 20$ time series, each with length $T = 50$, which ensures a consistent sample size and a fair comparison. Figure A8 shows the results in the settings with linear and nonlinear structural equation models and the Lorenz model.

We note that `DyCausal` attains significant speedups against TECDI, NGM, and NTS-NO (especially for the case with 80 nodes), which is attributed to our improved acyclic constraint. What is even more exciting is that `DyCausal` consistently achieves improved structural precision (lowest SHD) and competitive TPR and F1 under three causal models. This could be attributed to the fact that the reconstruction of complete time series is fragmented into the reconstructions of series

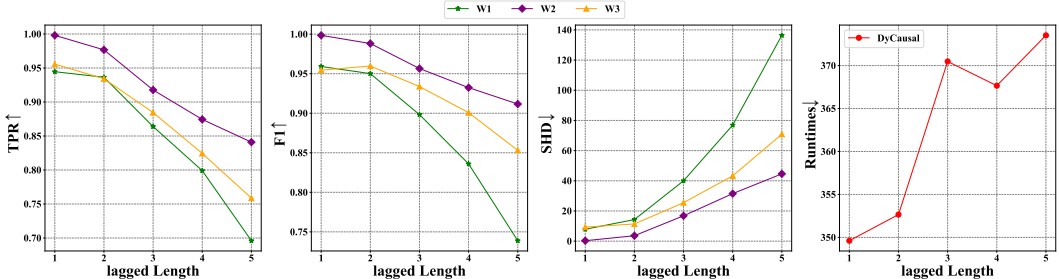

Figure A6: Results of `DyCausal` on datasets with maximum lag $\tau = \{1, 2, 3, 4, 5\}$.

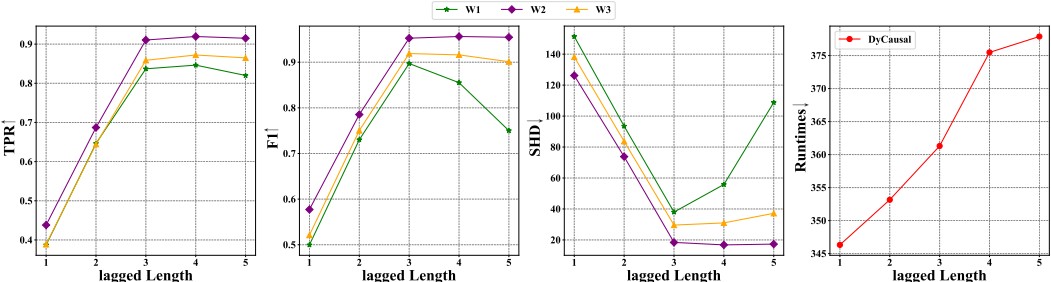

Figure A7: Results of `DyCausal` on datasets with maximum lag $\tau = 3$.

within sliding windows, which dilutes the negative effect of noise and improves the performance of `DyCausal`. In Figure A8, we only show SHD less than 1000 and runtimes less than 2000; results beyond this range lose competitiveness. Since the runtimes of TECDI on graphs with 80 nodes are more than 12 hours, we only report the results of TECDI on graphs with 10, 20, and 40 nodes. We report the results with variance in Table A4. We can observe that `DyCausal` achieves the best F1 and SHD in various models. In most settings, the TPR of `DyCausal` is also the highest. In addition, the result variance of `DyCausal` is very low, further demonstrating the robustness of `DyCausal` in various models.

## C.10 ABLATION FOR THE ACYCLIC CONSTRAINT IN NONLINEAR SETTINGS

To demonstrate the necessity of improving the log-det acyclic constraint, we introduce an ablation baseline `DyCausal` with $h_{log}$, which uses the original log-det acyclic constraint without scaling the matrix through the matrix norm. We compare `DyCausal` and `DyCausal` with $h_{log}$ on the causal graph with 10 nodes and nonlinear causality, where the maximum lag length $\tau = 2$. Figure A9 shows the ground-truth causal graph and the causal graphs estimated by `DyCausal` and `DyCausal` with $h_{log}$. Figure A10 shows the trajectories of $h_{norm}$ and $h_{log}$ indexed by the number of iterations.

We can see that `DyCausal` not only perfectly identifies the causal graph, but also its acyclic constraint value is effectively reduced as the iteration proceeds. In contrast, `DyCausal` with $h_{log}$ fails catastrophically. As expected, $h_{log}$ frequently exceeds the optimizable space during training, forcing the algorithm to roll back the model, reduce the learning rate, and retrain. As a result, `DyCausal` with $h_{log}$ runs over 60000 iterations, which is much more than the number of iterations `DyCausal` runs. In addition, due to the excessive reduction of the learning rate, the model cannot be effectively optimized to identify causal structures. We also conduct experiments on the causal graph with 10 nodes and dynamic settings, where the maximum lag is $\tau = 1$. Figure A11 shows the causal graph estimated by `DyCausal` and `DyCausal` with $h_{log}$ when $h_{log}$ runs successfully. Figure A12 shows the trajectories of $h_{norm}$ and $h_{log}$ indexed by the number of iterations.

The results show that the use of $h_{norm}$ can identify more accurate causal structures than that of $h_{log}$. Especially at both ends of the time series ($\mathbf{W}_1$ and $\mathbf{W}_{50}$), `DyCausal` almost perfectly identifies the causal graphs, while `DyCausal` with $h_{log}$ struggles with redundant and missing edges. The trajectories of $h_{norm}$ and $h_{log}$ indicate that $h_{norm}$ decreases more steadily early in training and

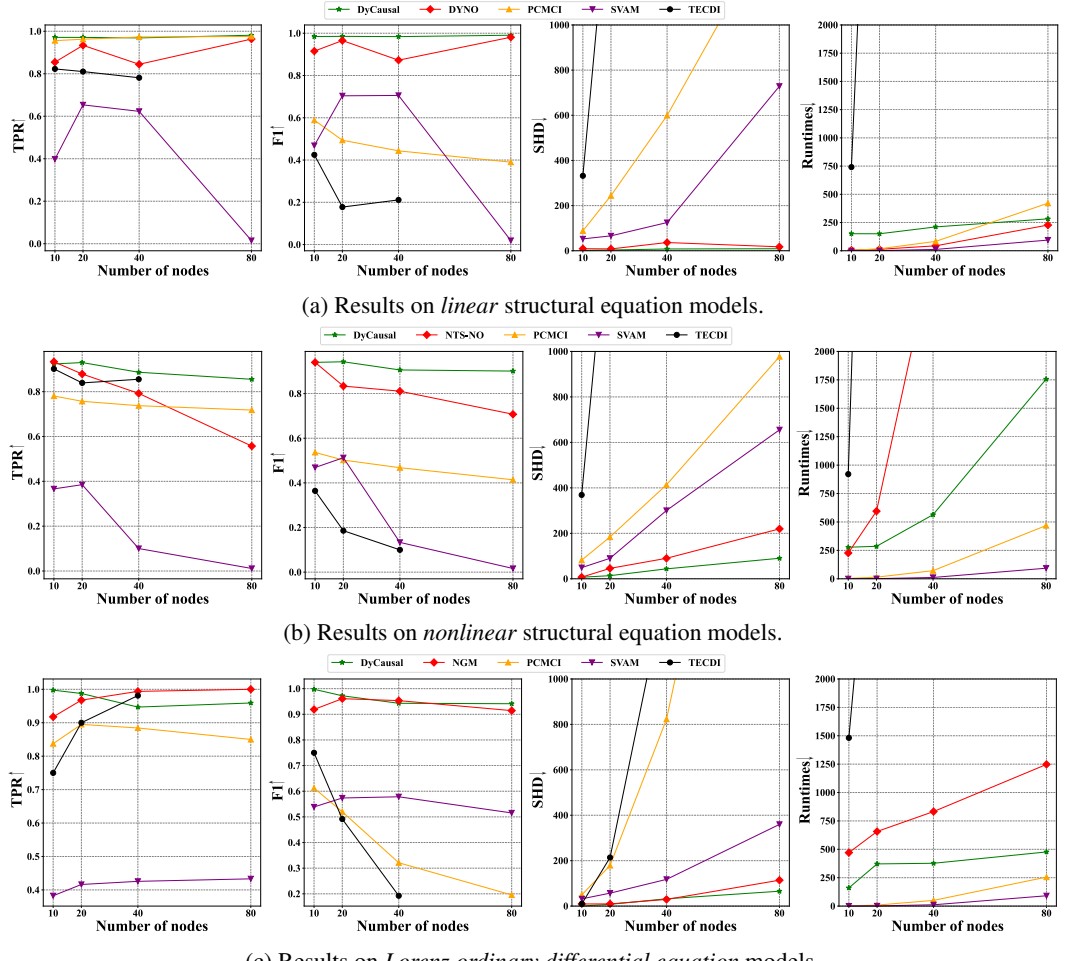

(a) Results on *linear* structural equation models.

(b) Results on *nonlinear* structural equation models.

(c) Results on *Lorenz ordinary differential equation* models.

Figure A8: Results on time series with static settings.

is closer to 0 at the end of training. The above results demonstrate that $h_{norm}$ outperforms $h_{log}$, whether in static or dynamic settings.

### C.11 SENSITIVITY ANALYSIS OF PENALTY COEFFICIENT

We conducted a sensitivity analysis on the penalty coefficient $\beta$ to prove that our parameter setting is reasonable. We run `DyCausal` on $N = 200$ time series with length $T = 50$ and number of variables $d = 50$. Table A5 reports the performance of `DyCausal` in $\beta = \{0.001, 0.005, 0.01, 0.05, 0.1\}$.

The results show that both too small and too large $\beta$ can lead to performance degradation, and `DyCausal` performs best when $\beta = 0.05$, which is consistent with our parameter setting.

### C.12 RESULTS ON NON-SMOOTH DYNAMIC CAUSALITY

In this paper, we assume that causal relationships change smoothly over time and use sliding Windows and coarse-grained approximations to learn dynamic causal relationships. However, we emphasize that `DyCausal` can adapt to non-smooth changing causal relationships. We discuss `DyCausal`'s behavior in two cases. We set causal matrix as $\mathbf{W}_t[i, j] = \mathbf{W}_0[i, j]\cos(\frac{t}{P}\frac{\pi}{E})$, where $P$ regulates the frequency of weight changes, and $E$ regulates the intensity of weight changes. In one case, we set $P = 4$, $E = 2$, window size $K = 3$ and sliding step size $S = 1$. In the other case, we set $P = 1$, $E = 2$, window size $K = 1$ and sliding step size $S = 1$. On $N = 200$ time series

Table A4: Results on time series with static causality. The boldfaced and underlined results respectively highlight the methods that perform the first and second best among all compared methods. We omit the percentage signs % of TPR and F1 to simplify the representation.

| | matrices | DyCausal | DYNO | PCMCI | SVAM |
|---|---|---|---|---|---|
| | | | linear structural equation model | | |
| | matrices | DyCausal | DYNO | PCMCI | SVAM |
| 10 nodes | TPR↑ | **97.13±2.42** | 85.46±6.95 | 95.59±1.83 | 39.79±22.34 |
| | F1↑ | **98.45±1.29** | 91.54±4.45 | 58.87±4.26 | 46.86±24.15 |
| | SHD↓ | **1.80±1.54** | 9.30±4.65 | 88.30±12.81 | 52.60±19.27 |
| 20 nodes | TPR↑ | **97.11±2.43** | 93.38±3.79 | 96.26±1.67 | 65.35±6.84 |
| | F1↑ | **98.52±2.43** | 96.54±2.06 | 49.37±4.94 | 70.40±5.55 |
| | SHD↓ | **3.40±2.84** | 8.20±4.92 | 244.10±42.06 | 66.10±13.60 |
| 40 nodes | TPR↑ | 96.87±1.67 | 84.45±2.79 | **97.31±0.77** | 62.30±2.78 |
| | F1↑ | **98.40±8.73** | 87.25±2.85 | 44.35±3.69 | 70.57±1.59 |
| | SHD↓ | **7.90±4.06** | 36.50±6.45 | 599.10±80.78 | 125.00±6.99 |
| 80 nodes | TPR↑ | **98.13±1.06** | 96.33±2.49 | 97.56±0.49 | 1.51±0.53 |
| | F1↑ | **99.06±0.54** | 98.12±1.32 | 39.01±3.10 | 1.93±0.68 |
| | SHD↓ | **9.00±5.35** | 17.60±12.02 | 1484.00±164.19 | 728.50±46.06 |
| | | | nonlinear structural equation model | | |
| | matrices | DyCausal | NTS-NO | PCMCI | SVAM |
| 10 nodes | TPR↑ | 92.35±3.12 | **93.33±5.49** | 78.05±4.99 | 36.63±7.04 |
| | F1↑ | **94.00±1.67** | 94.00±4.24 | 53.65±5.78 | 46.88±8.05 |
| | SHD↓ | **7.10±2.07** | 7.20±4.26 | 82.10±17.16 | 49.20±9.91 |
| 20 nodes | TPR↑ | **92.98±2.70** | 87.87±6.51 | 75.69±4.08 | 38.51±3.86 |
| | F1↑ | **94.26±2.73** | 83.37±8.12 | 50.21±4.29 | 51.27±4.37 |
| | SHD↓ | **13.80±6.35** | 45.90±30.49 | 183.70±22.29 | 90.10±10.45 |
| 40 nodes | TPR↑ | **88.66±3.32** | 79.23±4.97 | 73.74±2.27 | 10.02±16.11 |
| | F1↑ | **90.60±2.70** | 81.08±4.00 | 46.80±3.29 | 13.33±21.05 |
| | SHD↓ | **43.80±12.32** | 90.20±24.83 | 413.40±49.07 | 300.50±71.72 |
| 80 nodes | TPR↑ | **85.52±3.86** | 55.72±2.87 | 71.79±1.29 | 1.13±0.56 |
| | F1↑ | **90.09±2.35** | 70.75±2.71 | 41.40±1.79 | 1.64±0.79 |
| | SHD↓ | **90.40±21.72** | 219.67±21.95 | 977.30±48.88 | 655.40±30.14 |
| | | | ordinary differential equation model | | |
| | matrices | DyCausal | NGM | PCMCI | SVAM |
| 10 nodes | TPR↑ | **99.75±0.75** | 91.75±2.51 | 83.75±5.39 | 38.25±3.72 |
| | F1↑ | **99.75±0.50** | 91.90±2.16 | 61.23±3.77 | 53.87±3.84 |
| | SHD↓ | **0.20±0.60** | 9.30±2.93 | 49.70±5.60 | 31.90±3.48 |
| 20 nodes | TPR↑ | **98.75±1.25** | 96.75±1.39 | 89.50±3.63 | 41.63±2.68 |
| | F1↑ | **97.24±96.09** | 96.09±0.86 | 52.00±2.99 | 57.37±2.44 |
| | SHD↓ | **7.70±2.61** | 9.60±2.76 | 178.70±20.94 | 57.00±5.53 |
| 40 nodes | TPR↑ | 94.30±1.67 | **99.38±0.88** | 88.44±4.03 | 42.56±2.59 |
| | F1↑ | 94.30±1.67 | **95.34±1.29** | 32.11±2.05 | 57.84±2.38 |
| | SHD↓ | 32.20±9.22 | **29.30±8.26** | 823.10±48.14 | 117.60±8.79 |
| 80 nodes | TPR↑ | 95.91±1.26 | **100.00±0.00** | 85.00±2.74 | 43.31±1.76 |
| | F1↑ | **94.13±1.16** | 91.44±1.50 | 19.56±0.85 | 51.54±2.25 |
| | SHD↓ | **65.50±13.25** | 114.00±21.65 | 3148.30±94.59 | 359.70±31.79 |

with length $T = 10$ and number of variables $d = 50$, we validate the performance of DyCausal against non-smooth changes. The results are reported in Table A6:

It can be seen that in the first case, DyCausal accurately learned the causal graphs at both ends and in the middle of the time series. However, we find that the causality within the sliding window does not change at both ends and in the middle of the time series, which might be the reason why DyCausal accurately identified the causal graphs. Suppose that the causality changes from $\mathbf{W}$ to $\mathbf{W}'$ within the sliding window, where the sample size of the causality $\mathbf{W}$ is $a$ and the sample size of the causality $\mathbf{W}'$ is $b$. Observing the results, we find that the causality learned by DyCausal within this window is $\frac{a\mathbf{W}+b\mathbf{W}'}{a+b}$. We define the window with changing causality as a transition win-

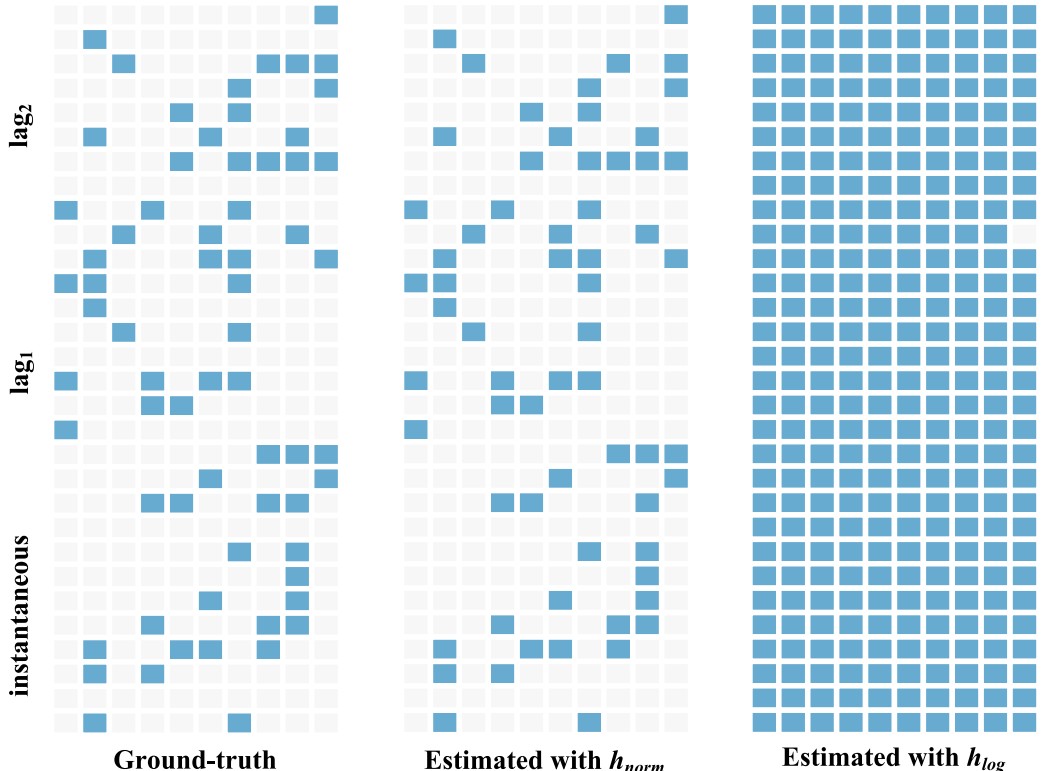

Figure A9: Results of DyCausal and DyCausal with $h_{log}$. The left is the ground-truth causal graph, the middle is the causal graph estimated by DyCausal, and the right is the causal graph estimated by DyCausal with $h_{log}$.

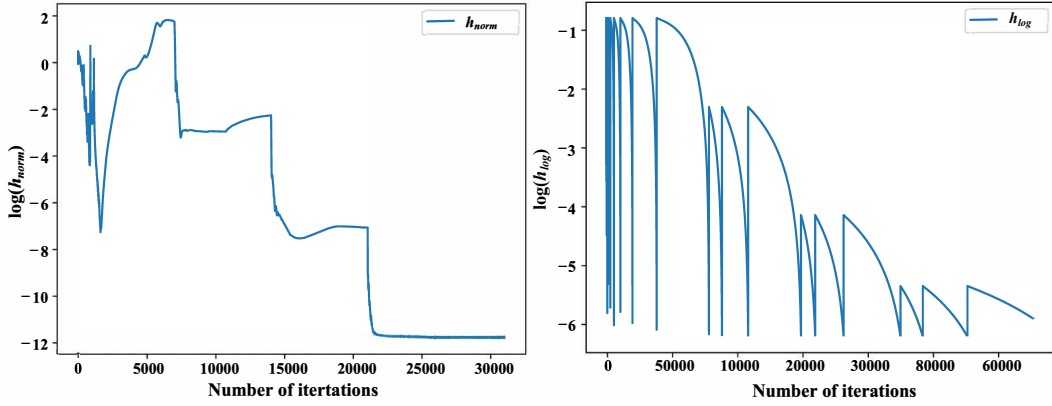

Figure A10: The trajectory of acyclic constraint values indexed by the number of iterations.

dow; otherwise, it is a stationary window. DyCausal can identify the changing trend of causality and accurately identify the causal graphs in the stationary window. In the second case, we set the window size to 1. Due to the small sample size within the window, DyCausal achieves a reduced performance, but still identified causal graphs with acceptable accuracy at each time step. Although the coarse-grained approximation no longer works when setting window size to 1, it is not usual to change the causality as frequently and strongly as in this experiment. More often, we can set the window size $K > 1$ as in the first case, identify the changing trend of the causality, and learn the accurate causal graphs in the stationary window.

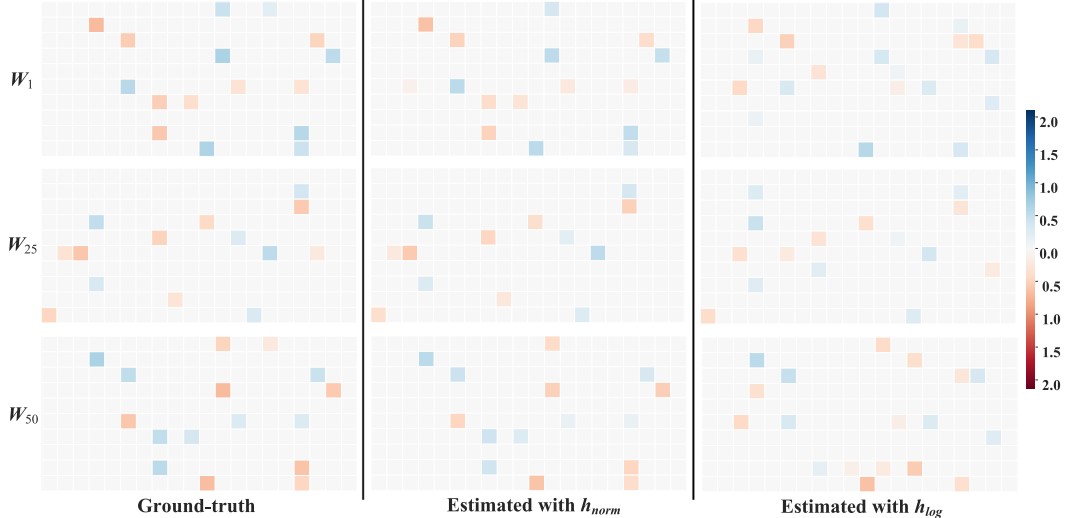

Figure A11: Results of `DyCausal` and `DyCausal` with $h_{log}$. The left is the ground-truth causal graphs, the middle is the causal graphs estimated by `DyCausal`, and the right is the causal graphs estimated by `DyCausal` with $h_{log}$.

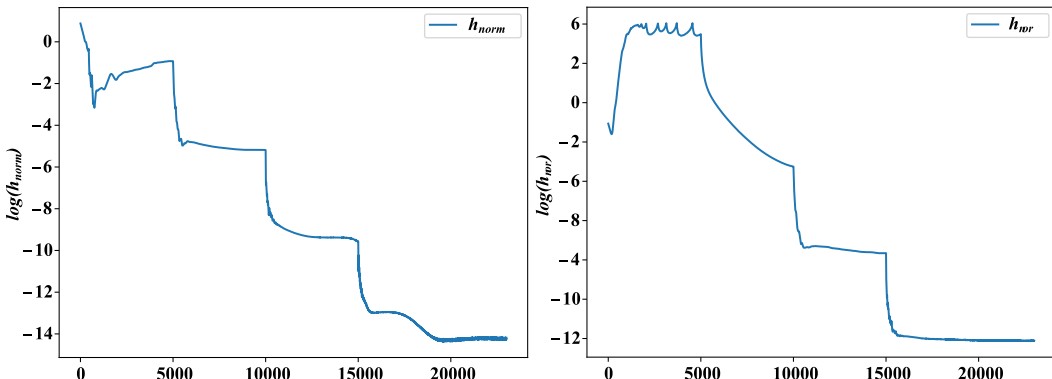

Figure A12: The trajectory of acyclic constraint values indexed by the number of iterations.

## C.13 RESULTS ON INSTANTANEOUS AND LAGGED CAUSALITY

To demonstrate `DyCausal`'s ability to, respectively, identify instantaneous and lagged causal relationships, we conducted experiments in dynamic settings. Specifically, we conduct `DyCausal` on $N = 200$ time series and set the length of series and the number of variables to $(T = 10, d = 20)$ and $(T = 50, d = 50)$. The results are reported in Table A7:

The results show that DyCausal accurately identifies the instantaneous and lagged causal relationships.

## C.14 RESULTS ON REAL-WORLD DATASETS

We also run `DyCausal` on four additional real-world datasets, NetSim (Smith et al., 2011), DREAM-3 (Prill et al., 2010), Phoenix (Hossain et al., 2024) and CausalRivers Stein et al. (2025), to prove its generalization and adaptability to real-world issues. In particular, we chose CUTS+ and NGM as baselines for comparison. The former performs second only to `DyCausal` on the CausalTime dataset, and the latter models ordinary differential equations to identify causality, which are often used in time series in the biological field. We record AUROC in Table A8 to estimate their performance.

Table A5: Results of sensitivity analysis on $\beta$.

|  |  | $\beta = 0.001$ | $\beta = 0.005$ | $\beta = 0.01$ | $\beta = 0.05$ | $\beta = 0.1$ |
|---|---|---|---|---|---|---|
| | TPR | 86.07 | _88.91_ | 77.09 | **93.65** | 55.95 |
| $\mathbf{W}_1$ | F1 | 66.46 | _77.36_ | 70.02 | **95.49** | 63.68 |
| | SHD | 99.0 | _51.5_ | 70.0 | **9.0** | 55.0 |
| | TPR | 92.87 | 95.78 | **98.95** | _98.93_ | 79.06 |
| $\mathbf{W}_{25}$ | F1 | 88.37 | 94.77 | _98.43_ | **99.46** | 86.83 |
| | SHD | 24.0 | 10.0 | _3.0_ | **1.0** | 63.0 |
| | TPR | 81.48 | _87.91_ | 79.65 | **95.09** | 48.66 |
| $\mathbf{W}_{50}$ | F1 | 76.20 | 74.52 | _77.36_ | **96.06** | 59.08 |
| | SHD | 54.0 | _59.5_ | _59.5_ | **8.0** | 63.0 |

Table A6: Results on non-smooth dynamic causality.

| | $P=4, E=2$ | | | $P=1, E=2$ | | | | | | | | | |
|---|---|---|---|---|---|---|---|---|---|---|---|---|---|
| | $\mathbf{W}_1$ | $\mathbf{W}_4$ | $\mathbf{W}_8$ | $\mathbf{W}_1$ | $\mathbf{W}_2$ | $\mathbf{W}_3$ | $\mathbf{W}_4$ | $\mathbf{W}_5$ | $\mathbf{W}_6$ | $\mathbf{W}_7$ | $\mathbf{W}_8$ | $\mathbf{W}_9$ | $\mathbf{W}_{10}$ |
| TPR | 93.11 | 87.92 | 91.21 | 91.92 | 97.06 | 86.87 | 94.12 | 86.87 | 91.18 | 87.88 | 90.20 | 89.90 | 95.10 |
| F1 | 92.78 | 97.21 | 90.93 | 88.78 | 92.52 | 82.30 | 90.57 | 83.90 | 88.15 | 85.29 | 88.46 | 84.76 | 89.81 |
| SHD | 13.7 | 5.3 | 17.2 | 23.00 | 16.00 | 37.00 | 20.00 | 33.00 | 25.00 | 30.00 | 24.00 | 32.00 | 22.00 |

We observe that CUTS+ achieves the lowest AUROC on NetSim. This may be because the data generation process of NetSim is more consistent with the ODE model. Therefore, NGM achieves better performance than CUTS+ on NetSim. Both DREAM and Phoenix are datasets about gene expression levels, where DREAM contains 100 genes and Phoenix contains 11165 genes. Due to the large scale and the lack of a verifiable causal network, we screen 34 transcription factors based on the gene regulatory network provided by Phoenix and take the regulatory network of these transcription factors as the target of causal discovery. For CausalRivers, we use the default sub-dataset Random 5. This sub-dataset contains 5 randomly selected river nodes and provides the flow time series and causal graph of these nodes. The results show that `DyCausal` achieves significantly higher AUROC than the baselines on all four datasets. This is attributed to flexible adjustment of the reconstruction model to adapt to different datasets.

## D    IDENTIFIABILITY OF `DyCausal`

Identifiability is the most fundamental issue in causal discovery, and in this section, we discuss the identifiability of `DyCausal`. Looking back at the process of learning causal structures, we reconstruct the data in the window $[t, t + K]$ to optimize the causal structures at the time step $t$. Although this process slightly violates the dynamic causality (causality at time $t$ is only responsible for generating observations at time $t$), we assume that $K$ is small enough and that causality is approximately invariant within the interval $[t, t + K]$. Therefore, identifying dynamic causality across the entire time series is broken down into identifying approximately static causality within multiple sufficiently small intervals. According to the identifiability results in (Shimizu et al., 2006; Peters & Bühlmann, 2014; Hoyer et al., 2008), when the sample size tends to infinity, no other DAG can generate the same distribution. Therefore, when observed data within the interval $[t, t + K]$ are generated by the structural equation model defined in Eq. (1), and the observed sample size of the time series is sufficient, we can identify the Markov equivalence class of the causal graph and restore the coefficients of the causal matrix $\mathbf{W}_t$ at time $t$. Furthermore, based on the identified coarse-grained causal matrices, we restore fine-grained causal matrices through linear interpolation. According to Eq. (5), if the dynamic causal matrix is a continuous function of the time index, we can approximate the causal matrix at each time step by linear interpolation within a sufficiently small interval $[t, t + S]$. In summary, dynamic causality is identifiable for `DyCausal`.

## E    THE USE OF LARGE LANGUAGE MODELS

We promise that we did not use large language models (LLMs) in this work.

Table A7: Results of instantaneous and lagged causality.

| | results of instantaneous causal graphs | | | | | | | | | | | | |
|---|---|---|---|---|---|---|---|---|---|---|---|---|---|
| | $T=50, d=50$ | | | $T=10, d=20$ | | | | | | | | | |
| | $\mathbf{W}_1$ | $\mathbf{W}_{25}$ | $\mathbf{W}_{50}$ | $\mathbf{W}_1$ | $\mathbf{W}_2$ | $\mathbf{W}_3$ | $\mathbf{W}_4$ | $\mathbf{W}_5$ | $\mathbf{W}_6$ | $\mathbf{W}_7$ | $\mathbf{W}_8$ | $\mathbf{W}_9$ | $\mathbf{W}_{10}$ |
| TPR | 92.03 | 97.34 | 91.52 | 86.91 | 83.03 | 86.12 | 81.39 | 91.29 | 91.29 | 81.56 | 85.19 | 83.60 | 83.64 |
| F1 | 94.45 | 98.24 | 93.84 | 89.27 | 89.53 | 91.80 | 89.43 | 93.62 | 93.88 | 89.03 | 90.19 | 89.59 | 86.47 |
| SHD | 7.2 | 2.2 | 6.8 | 6.2 | 9.8 | 8.5 | 8.7 | 2.7 | 2.8 | 8.8 | 9.5 | 10.3 | 8.0 |
| | results of lagged causal graphs | | | | | | | | | | | | |
| | $T=50, d=50$ | | | $T=10, d=20$ | | | | | | | | | |
| | $\mathbf{W}_1$ | $\mathbf{W}_{25}$ | $\mathbf{W}_{50}$ | $\mathbf{W}_1$ | $\mathbf{W}_2$ | $\mathbf{W}_3$ | $\mathbf{W}_4$ | $\mathbf{W}_5$ | $\mathbf{W}_6$ | $\mathbf{W}_7$ | $\mathbf{W}_8$ | $\mathbf{W}_9$ | $\mathbf{W}_{10}$ |
| TPR | 94.38 | 96.97 | 92.92 | 90.81 | 86.14 | 90.85 | 84.25 | 93.16 | 93.13 | 84.02 | 87.56 | 86.67 | 90.28 |
| F1 | 96.67 | 98.30 | 95.50 | 93.50 | 91.57 | 94.79 | 91.12 | 95.75 | 96.32 | 90.50 | 92.29 | 91.84 | 92.43 |
| SHD | 6.0 | 3.2 | 7.6 | 4.8 | 8.4 | 5.9 | 7.5 | 2.5 | 2.1 | 8.3 | 8.7 | 8.3 | 5.7 |

Table A8: AUROC on NetSim, DREAM-3 and Phoenix.

| | NetSim ($d=15$) | DREAM ($d=100$) | Phoenix ($d=34$) | CausalRivers ($d=5$) |
|---|---|---|---|---|
| DyCausal | **0.9391±0.0075** | **0.6806±0.0197** | **0.5472±0.0185** | **0.7214±0.0405** |
| CUTS+ | 0.7058±0.0071 | 0.6229±0.0138 | 0.5104±0.0151 | 0.6072±0.0627 |
| NGM | 0.7499±0.0020 | 0.5629±0.0020 | 0.4809±0.0083 | 0.4905±0.0526 |

