# OpenReview forum: "Coarse-to-Fine Learning of Dynamic Causal Structures"
_ICLR.cc/2026/Conference — ICLR 2026 Poster_

### Official Review · Reviewer_YHJh · 2025-10-15

**Soundness:** 3
**Presentation:** 3
**Contribution:** 3
**Rating:** 6
**Confidence:** 3

**Summary:**

This paper introduces a novel method for time-series causal discovery, specifically designed to handle dynamically changing causal relationships. This is a timely and important problem that has received relatively little attention in the literature. The proposed approach is well-motivated, and the paper is clearly written and supported by extensive empirical evaluations.

While the work is promising, several points regarding the experimental validation, comparison to related work, and clarity of the evaluation protocol require attention. Addressing these points would strengthen the paper and allow me to raise my rating further.

**Strengths:**

- **Clarity and Presentation:** The paper is well-written and mostly easy to follow.
- **Novelty and Timeliness:** The paper addresses an important and under-explored problem in causal discovery: learning dynamic causal graphs from time-series data. This represents a valuable and timely contribution.
- **Empirical Evaluation:** The authors provide an extensive set of experiments on both synthetic and real-world datasets, which helps to demonstrate the method's effectiveness across various settings.

**Weaknesses:**

### **Major Weaknesses**

1. **Clarity on Real-World Dataset Dynamics and Evaluation:** The justification for using certain real-world datasets (e.g., `NetSim` or `CausalTime`) to evaluate a *dynamic* causal discovery method is unclear. To the best of my understanding, these datasets have a static ground truth graph (as they rely on some synthetic components).
    - The authors should clarify why a dynamically changing graph is expected or found in these cases. If there is no prior information on dynamics, this should be explicitly stated, as it impacts the interpretation of the results.
    - To further validate the method's ability to handle known dynamics, the authors are strongly encouraged to evaluate on benchmarks like **CausalRivers** [6]. This dataset's assumption of a fixed graph structure with dynamically changing edge weights perfectly matches the problem setting of this paper and would provide more convincing real-world evidence.

2. **Usage of $h_{norm}$:** While Eq. 7  suggests that the norm is used as a constraint, Algorithm 1 shows that it is simply added as a loss term. This suggests that it is not enforced that $h_{norm}$ is exactly 0, but rather it is only pushed close to 0 to minimize the loss. As the DAG structure is only enforced at $h_{norm} = 0$, how is it guaranteed that the solution is really a DAG? Please clarify this point.

3. **Ambiguity in Evaluation Protocol:** How are the performance metrics in Table 1 calculated for dynamic graphs? For instance, are the predicted graphs over time collapsed into a single summary graph for comparison with a static ground truth? How are baseline methods that produce only a single, static graph treated in this evaluation? This information is essential for interpreting the results correctly.

4. **Lack of Architectural Comparison and Positioning:** The paper would be strengthened by a more detailed comparison with related convolutional approaches such as NTS-NOTEARS, TCDF, and other recent works (e.g., [4, 5]). A discussion of the architectural differences and how they relate to the modeling of temporal dependencies would help readers better understand the unique contributions and advantages of the proposed method.

5. **Need for Granular Performance Analysis:** The paper would benefit from a more fine-grained analysis of the model's performance. Specifically, providing separate recovery scores for **instantaneous** and **lagged** causal links would offer deeper insights into the method's strengths and weaknesses.
7. **Unexplained Superiority on Static Graphs (Appendix A8):** A key result that requires further explanation is the proposed method's superior performance even on static graphs. This is counterintuitive and could suggest suboptimal hyperparameter tuning for the baseline methods. A more thorough investigation or discussion is needed to justify this finding and rule out experimental artifacts.



### **Minor Weaknesses**

- **Assumption on Graph Structure:** The method appears to assume a fixed graph structure with dynamically changing edge weights. While not a significant weakness, this should be made clear earlier in the paper, as the method does not address the problem of a changing graph *structure* (i.e., addition/removal of edges).
- **Smoothness Assumption:** The paper could benefit from a brief discussion on the limitations of the smoothness assumption for the function `F`. What happens in natural systems with phase transitions or abrupt changes (e.g., freezing at 0°C), where this assumption is violated? A small synthetic experiment on this (e.g., simply drawing coefficients from a uniform distribution) would be very interesting.
- **Literature Review:** The background section mentions two groups of CD methods, which may be an oversimplification. The review could be extended to better position the work by including other major families of methods, such as Granger Causality and constraint-based/noise-based approaches [1].
- **Related Work:** A recent paper at ICLR, on Meta Causal Graphs [2], appears closely related and should be discussed.
- **Terminology:**
    - **L49 "Faithful":** The use of the word "faithful" could be confused with the Causal Faithfulness assumption, a standard term in the causal inference literature. Rephrasing is recommended.
    - **"Significant":** The word "significant" is often used without statistical validation. Please ensure its use is appropriate or rephrase to be more descriptive (e.g., "substantial," "large").
    - **SEM vs. SCM:** The paper uses the term Structural Equation Model (SEM). To avoid confusion with the large body of work in social sciences also using this term, consider using Structural Causal Model (SCM) instead, as suggested by [3].
- **Clarity and Typos:**
    - **Table 1:** Typo `CUT+` should be `Cuts+`.
    - **Figures:** Many figures use labels like `W1`, `W2`, `W3` without a clear definition in the caption or in the legend. Adding a brief explanation in the figure captions would improve comprehension.
    - **Precision Metric:** The precision values in the tables appear to be between 0 and 100 but are reported without a decimal point (e.g., "95" instead of "0.95"). This should be clarified in the table captions.



¹  https://www.jair.org/index.php/jair/article/view/13428/26917

² https://openreview.net/forum?id=J9VogDTa1W

³ https://library.oapen.org/bitstream/id/056a11be-ce3a-44b9-8987-a6c68fce8d9b/11283.pdf

⁴ https://arxiv.org/abs/2408.08023

⁵ https://arxiv.org/html/2404.01466v1

⁶ https://openreview.net/forum?id=wmV4cIbgl6

**Questions:**

Answering these questions in the paper or rebuttal would greatly improve clarity:

1. **Robustness to Non-Cyclic Dynamics:** What is the expected behavior of the model if the underlying causal dynamics are not smooth or cyclic, but instead feature abrupt changes or non-periodic patterns?
2. **Acyclicity Constraint:** Could the acyclicity constraint be relaxed or removed? If so, what would be the expected impact on performance?
3. **Performance on Static Graphs:**  Can you give a rational why your method is typically even better than other methods in the static case? (A8). This suggests some Hyperparameter problems.
4. **Interpolation vs. Per-Step Estimation:** Do you have to perform linear interpolation, or is this something you simply do for computational efficiency? If this is the case, how do results change if the Causal Graph is estimated at every time step?
5. **Dynamics of Instantaneous Links:** There appears to be a contradiction between Eq. 2 (which suggests instantaneous links are static) and Section 5.1 (where they seem to be varied in experiments). Am I misunderstanding something?

---

> ### Author Response · Authors · 2025-11-19
>
> We sincerely thank the reviewer YHJh for their positive rating, and for their thoughtful questions and concerns. Below, we address each point constructively:
>
> **Q1: Robustness to non-cyclic dynamics**
>
> **A1: DyCausal can handle non-smooth dynamic causality.** We discuss DyCausal’s behavior in two cases. We set the causal matrix as $W_t[i,j]=W_0[i,j]cos(\lfloor\frac{t}{P}\rfloor\frac{\pi}{E})$, where $P$ regulates the frequency of weight changes, and $E$ regulates the intensity of weight changes. In one case, we set $P=4$, $E=2$, window size $K=3$ and sliding step size $S=1$. On $N=200$ time series with length $T=10$ and number of variables $d=50$, we valid the performance of DyCausal against non-smooth changes. The results are as follows:
>
> | |TPR |F1|SHD|
> |-|-|-|-|
> $W_1$|93.11|92.78|13.7|
> $W_4$|87.92|97.21|5.3|
> $W_8$|91.21|90.93|17.2|
>
> It can be seen that DyCausal accurately learned the causal graphs at both ends and in the middle of the time series. However, we find that the causality within the sliding window do not change at both ends and in the middle of the time series, which might be the reason why DyCausal accurately identified the causal graphs. Suppose the causality changes from $W$ to $W'$ within the sliding window, where the sample size of the causality $W$ is $a$ and the sample size of the causality $W'$ is $b$. By observing the results, we find that the causality learned by DyCausal within this window is $(aW+bW')/(a+b)$. If the steps with the window where the causality changes are called transition steps and the steps with the window where the causality does not change are called stop steps, then DyCausal can identify the changing trend of the causality and accurately identify the causal graphs at the stop step.
> In the other case, we set $P=1$, $E=2$, window size $K=1$ and sliding step size $S=1$. The results are as follows:
>
> | |$W_1$|$W_2$|$W_3$|$W_4$|$W_5$|$W_6$|$W_7$|$W_8$|$W_9$|$W_{10}$|
> |-|-|-|-|-|-|-|-|-|-|-|
> TPR|91.92|97.06|86.87|94.12|86.87|91.18|87.88|90.20|89.90|95.10|
> F1|88.78|92.52|82.30|90.57|83.90|88.15|85.29|88.46|84.76|89.81|
> SHD|23.00|16.00|37.00|20.00|33.00|25.00|30.00|24.00|32.00|22.00|
>
> Facing frequent and non-smooth changes in the causality, we set the window size to 1. Due to the small sample size within the window of size 1, DyCausal achieved reduced performance but still identified causal graphs with acceptable accuracy at each time step. Although setting the window size to 1 causes the coarse-grained approximation to no longer work, it is not common for the causality to change at each time step. Most of the time, we can set the window size $K>1$ as in the first case, identify the changing trend of the causality, and learn the accurate causal graphs at the stop points.
>
> **Q2: Acyclicity constraint**
>
> **A2: Acyclic constraints can be relaxed or removed.** In fact, we have included an example of removing acyclic constraint in the experiments. As shown in Eq.(9), we remove the acyclic constraint since the ODE causal model does not include instantaneous causal effects. The results in Table A3 indicate that DyCausal without acyclic constraints can still accurately learn causal structures. We emphasize a principle that has always been adhered to, that is, we need to constrain the acyclic instantaneous causality if it exists, and we have already explained the reasons for adhering to this principle in line 140 of this paper. On $N=200$ time series with the length $T=50$ and the number of variables $d=50$, we conducted experiments on the dynamic settings without instantaneous causal relationships.
>
> | |TPR |F1|SHD|
> |-|-|-|-|
> $W_1$|98.78|99.38|1.2|
> $W_{25}$|100.00|100.00|0.0|
> $W_{50}$|97.93|98.54|2.7|
>
> The results show that DyCausal can even identify more accurate causal structures without considering the acyclic constraints.

---

> ### Author Response · Authors · 2025-11-19
>
> **Q3: Performance on static graphs**
>
> **A3: The reason why DyCausal is typically even better than other methods is that it uses a sliding window to learn the same causal structure at multiple time steps.** This process is similar to ensemble learning, and summarizing the causal graphs learned from multiple time steps greatly reduces the probability of the model making mistakes and improves the accuracy of causal discovery. We conduct a simple experiment to prove this point. We set the sliding window size to be consistent with the time series length, which means that DyCausal can only learn one causal graph from the entire time series. On $N=20$ time series with the length $T=50$ and the number of variables $d=40$, we conduct experiments on the linear, nonlinear and ODE model.
>
> |linear|TPR|F1|SHD|
> |-|-|-|-|
> DyCausal|97.80|98.42|7.3
> DyCausal only one graph|96.18|98.03|9.2
>
> |nonlinear|TPR|F1|SHD|
> |-|-|-|-|
> DyCausal|89.09|91.48|39.1|
> DyCausal only one graph|81.64|89.14|48.6
>
> |ODE|TPR|F1|SHD|
> |-|-|-|-|
> DyCausal|95.74|93.53|35.2|
> DyCausal only one graph|85.01|87.67|49.6|
>
> The results show a decrease in the performance of DyCausal, which indicates that our idea is correct. In addition, the hyperparameters we set fully exploited the performance of the baselines. We compared the results provided by DYNOTEARS and NFT-NOTEARS in their papers. When the maximum lag is 2 and the edge density is 2, the results we reported in Table A3 are consistent with those reported in Figure 3 of the paper of DYNOTEARS and Figure 6 of the paper of NTS-NOTEARS.
>
> **Q4: Interpolation vs. Per-step estimation**
>
> **A4: Linear interpolation is necessary for large-scale causal graphs and long time series.** Linear interpolation is proposed to cooperate with the strategy of learning causal structures in the sliding window. The sliding window can integrate the data within the window to learn causal structures with a larger sample size (which is related to the window size). In addition, the sliding window can coarse-grained traverse the time series to improve the computational efficiency (this is related to the sliding step size). Linear interpolation enables the coarse-grained causal matrices learned by the window to jointly participate in training; otherwise, each matrix can only be trained in isolation based on the data within the corresponding window. Our ablation experiments in Appendix C.2 also demonstrate that linear interpolation not only improves efficiency but also enhances performance. On $N=200$ time series with length 50 and 200 nodes, we estimate the causal graph at each time step under two settings. One sets the window size $K=4$ and the sliding step $S=1$. The other sets both the window size and the sliding step to one.
>
> | | |TPR|F1|SHD|runtime(ms)|
> |-|-|-|-|-|-|
> $W_1$|DyCausal|91.56|92.73|63|2926|
> $W_1$|K=4,S=1|90.54|90.95|72|3527|
> $W_1$|K=1,S=1|83.33|20.48|2485|1970|
> $W_{25}$|DyCausal|96.70|98.32|13|2926|
> $W_{25}$|K=4,S=1|95.47|9768|18|3527|
> $W_{25}$|K=1,S=1|89.22|30.22|1644|1970|
> $W_{50}$|DyCausal|94.18|94.18|46|2926|
> $W_{50}$|K=4,S=1|90.09|92.59|64|3527|
> $W_{50}$|K=1,S=1|85.68|20.76|2512|1970|
>
> The results show that DyCausal runs for a longer time in the former setting. That is because the model encodes more matrices and calculates the acyclic constraint term more times. DyCausal achieves reduced results in the latter setting. This is because the sample size within a window of size 1 is too small to support the model in learning causal graphs with 200 nodes.
>
> **Q5: Dynamics of instantaneous links**
>
> **A5: Eq.(2) does not conflict with the experimental settings in Section 5.1.** I think you might misunderstand Eq.(2), which requires that the weighted adjacency matrix W_t includes instantaneous and lagged causality changes over time, and that the instantaneous causality $W^{ins}_t$ is a DAG, but it does not require the instantaneous causality being static.

---

> ### Author Response · Authors · 2025-11-19
>
> **W1: Clarity on real-world dataset dynamics and evaluation**
>
> **A6: We run DyCausal on datasets NetSim and CausalTime to verify whether there are dynamic changes in the intensity of causal relationships on these datasets.** As you said, both the NetSim and CausalTime datasets rely on some models to simulate real data. Take CausalTime as an example, DyCausal uses deep models to fit real data, thereby generating synthetic data with a distribution similar to that of real data. However, it remains unknown whether there are dynamic changes in the real data generation model and the real causal relationship. Therefore, we applied DyCausal to CausalTime and found that the intensity of the causal relationships between some variable pairs in the sub-dataset traffic varies. Such a discovery is meaningful. According to the descriptions of sub-datasets by CausalTime, the traffic sub-dataset records the flow changes at traffic nodes within a day. The discovery in Figure 5 implies that the causal relationships between the traffic nodes have different patterns during the day and at night. We also conduct experiments on the benchmark CausalRivers. We use the interface provided by CausalRivers to sample subgraphs and time series with five nodes.
>
> | |AUROC|
> |-|-|
> DyCausal|0.7142|
> CUTS+|0.6072|
> NGM|0.5357|
>
> The results show that DyCausal outperforms the comparison methods. This may be because DyCausal is able to capture the weights of the dynamics.
>
> **W2: Usage of $h_{norm}$**
>
> **A7: Pruning ensures that the algorithm obtains a DAG.** As in line 7 of Algorithm 1, after obtaining the weight matrix, we prune the edges with small weights in the matrix based on the threshold, which ensures that the algorithm yields a DAG. We agree with your view that whether it is augmented Lagrangian optimization or central path optimization, the acyclic constraint term can only approach 0 rather than equal 0. However, an acyclic constraint term close to 0 means that the weights of the edges that make up the loop in the matrix will be very small. Therefore, we can easily prune the edges participating in the loop in the matrix using the threshold and obtain a DAG.
>
> **W3: Ambiguity in evaluation protocol**
>
> **A8: We clarify the evaluation strategy for the results in Table 1.** Since the CausalTime dataset only provides a single real causal graph, for the baselines of identifying static causal graphs, we directly run these baselines and compare their estimated graphs with the real causal graphs to obtain the evaluation results. For DyCausal, since it estimates the causal graph at multiple time steps, we first take the union of the estimated graphs to obtain a summary graph, and then compare the summary graph with the real causal graph to obtain the evaluation index. Our strategy follows an obvious principle: if there is a clear causal relationship between a pair of variables at a certain moment (which means that the causal relationship between the pair of variables has a stronger causal influence at that moment and a weaker one at other moments), we should retain this causal relationship.
>
> **W4: Lack of architectural comparison and positioning**
>
> **A9: Our architecture outperforms existing architectures in learning fully dynamic causal relationships.** Specifically, we use convolutional networks and parallel encoding networks to encode the causal matrices $W_t$, which is different from existing methods of directly extracting the $W$ from the model parameters. In addition, we introduce a coarse-grained approximation strategy of dynamic causal structure, which is not forcibly added but naturally fits with the sliding window. The extensive experimental results we provide all demonstrate the superiority of the above causal structure learning strategies. In the face of static causality, the sliding window of DyCausal is similar to ensemble learning and aggregates the causal graphs learned from multiple time steps, which greatly reduces the probability of the model making mistakes and improves the accuracy of causal discovery. Our experiment in Q1 proved this view. In addition, our improved acyclic constraint addresses an undeniable defect faced by $h_{log}$, which can only be optimized in a limited space. Furthermore, we demonstrated that $h_{norm}$ conforms to the stability criterion, while $h_{log}$ has previously been proven to be unstable. For a fair comparison, we also directly replace DAGMA's $h_{log}$ with $h_{norm}$ and conduct experiments on non-temporal data.
>
> | |TPR|FDR|SHD|
> |-|-|-|-|
> $h_{norm}$|84.10|6.82|22.4|
> $h_{log}$|78.90|11.80|29.7|
>
> Both Appendix C.10 and the above results indicate that the performance of $h_{norm}$ is superior to that of $h_{log}$.

---

> ### Author Response · Authors · 2025-11-19
>
> **W5: Need for granular performance analysis**
>
> A10: On $N=200$ time series with the length $T=50$ and the number of variables $d=50$, we conduct DyCausal in dynamic settings and report the results of instantaneous and lagged causal relationships.
>
> | | |TPR |F1|SHD|
> |-|-|-|-|-|
> $W_1$|instantaneous|92.03|94.45|7.2|
> $W_1$|lagged|94.38|96.67|6.0|
> $W_{25}$|instantaneous|97.34|98.24|2.2|
> $W_{25}$|lagged|96.97|98.30|3.2|
> $W_{50}$|instantaneous|91.52|93.84|6.8|
> $W_{50}$|lagged|92.92|95.50|7.6|
>
> The results show that DyCausal accurately identifies the instantaneous and lagged causal relationships.
>
> **W: Assumption on graph structure**
>
> **A11: DyCausal is indeed learning dynamic causal structures.** When generating synthetic data, we first sample an initial causal graph and sample the edge weights. Subsequently, we used Eq.(A4) to adjust the edge weights to change dynamically over time. Finally, we introduce a threshold to prune the edges with small weights at each time step, generating the final causal matrices, where both the increase and decrease of variable weights and the appearance and disappearance of edges exist).

---

> ### Comment · Reviewer_YHJh · 2025-11-26
>
> **I thank the reviewers for their extensive rebuttal.
> Most of my concerns were adequately addressed.**
>
> I have one remaining point concerning the CausalRivers experiments:
> Did you use one of the default datasets? If yes, how does your method compare to other results reported in the leaderboard?
> I was unable to find any additional information in the manuscript.
>
> Alternatively, there is another benchmark that focuses explicitly on dynamic causal graphs [1].
> I understand, however, that this is likely outside the scope of the remaining time for the rebuttal.
>
> [1] CausalDynamics: A large-scale benchmark for structural discovery of dynamical causal models, https://arxiv.org/abs/2505.16620

---

> ### Author Response · Authors · 2025-11-27
>
> Thank you once again for your valuable feedback. We are delighted to know that our response has addressed most of your concerns. Below, we address your remaining concerns:
>
> **Q1: Comparison in the leaderboard**
>
> **A1: We used the default dataset provided by CausalRivers.** Specifically, we used Random 5 in the default datasets. DyCausal outperforms most of the baselines in the leaderboard provided by CausalRivers, second only to VAR and CDMI and close to VARLiNGAM. This may be because VAR and CDMI identify causal relationships from all observations in the past. Unlike VAR and CDMI, DyCausal takes into account the maximum lag and identifies causal relationships only from the past few time steps. However, this does not mean that the limited lag causal relationship is unreasonable. In contrast, limited lag causal relationships are common. Compared with the causal discovery algorithms that also take into account the maximum lag in the leaderboard, DyCausal achieves satisfactory results, outperforms NOTEARS,PCMCI and CP, and is close to VARLiNGAM.
>
> **Q1: About CausalDynamics**
>
> **A1: The benchmark CausalDynamics provides static causal graphs.** We had focused on CausalDynamics. In fact, we found that CausalDynamics deliberately distinguishes between dynamic causal graphs and dynamical causal models. For example, Eq. (1) of CausalDynamics defines the ODE or SDE model as the structural dynamical causal model. However, we all know that the causal graph of the ODE or SDE model is static (We had provided static causal discovery results of DyCausal on Lorenz96 models). In my opinion, the dynamical causal model drives the system evolution with some elements as the motive force. For example, time is the driving force that drives the evolution of variables in ODE or SDE models. However, dynamic causal graphs refer to causal relationships enhancing, weakening, appearing and disappearing. Therefore, we believe that the causal graph provided by CausalDynamics remains static.
>
> We are incorporating our discussions into the revision to the best of our ability, including the results in CausalDynamics, and believe these discussions will significantly improve the paper.

---

> > ### Comment · Reviewer_YHJh · 2025-11-27
> >
> > Thank you for specifying these details.
> >
> > While I am not sure if I share the authors' confidence that the results are satisfactory, I also believe there may be many issues at work, as the data likely has other complications beyond its changing causal graph.
> >
> > In any case, I think it would be interesting to discuss these results as it could lead the direction for future work.
> >
> > Thank you for your efforts. I will keep my current score as the final assessment.

---

> > > ### Author Response · Authors · 2025-11-27
> > >
> > > Thank you for your thoughtful feedback and for your final positive assessment. We have incorporated your suggestions and crucial results from our discussion into the revisions. It is believed that these suggestions and supplements have significantly improved this paper and strengthened our confidence in the results. In the future, we will continue to focus on dynamic causal discovery, especially real benchmark datasets with dynamic causal structures. We believe that dynamic causal discovery provides a solid foundation and practical solutions for causal learning in more complex scenarios.

---

### Official Review · Reviewer_QNBN · 2025-10-29

**Soundness:** 3
**Presentation:** 3
**Contribution:** 2
**Rating:** 4
**Confidence:** 5

**Summary:**

This paper proposes a coarse-to-fine causal discovery method to address fully dynamic causality. They leverage CNN to capture patterns within coarse-grained time windows, and apply linear interpolation to refine causal structures. The authors tested DyCausal's performance and compared it with previous methods on both synthetic and real-world datasets.

**Strengths:**

1. The authors addressed the limitations of previous methods by considering a fully dynamic causality.

2. They offered clear motivation for their DyCausal algorithm and provided a new form of acyclic constraint.

3. The authors tested DyCausal's performance and compared it with previous methods.

**Weaknesses:**

1. The proposed refined dynamic causal graph interpolation is based on the assumption that the causal relationship changes smoothly and follows a linear law, which is a very strong prior assumption.

2. The acyclicity constraint in the question appears to be a variant of the DAGMA constraint, and I am skeptical as to whether it is sufficient to be considered an innovative point.

3. Learning causal relationships requires theoretical guarantees of identifiability. The proposed CNN method employs an encoder and decoder to infer causal structures, treating it as an unsupervised sequence reconstruction task, which appears to lack theoretical support.

**Questions:**

1. There seems to be a lack of sensitivity analysis of penalty coefficient $\beta$, decay coefficient $\mu$, and decay rate $\gamma$.

2. The model uses a sliding window approach for causal discovery. The question is whether the causal graphs obtained at the intersection of two windows are consistent. For example, is there a large difference in the changes between $W_{t+S}$ and $W_{t+S+1}$? Are they smooth?

3. The experimental results on CausalTime do not appear to include some of the latest state-of-the-art methods, such as JRNGC.

4. Are there any experiments on a real-world dataset with large-scale nodes?

5. As far as I know, DYNOTEARS and NTS-NOTEARS are both static causal discovery methods. How do they obtain the results on the dynamic dataset?

[1] Zhou, W., Bai, S., Yu, S., Zhao, Q., & Chen, B. (2024). Jacobian regularizer-based neural granger causality. arXiv preprint arXiv:2405.08779.

---

> ### Author Response · Authors · 2025-11-19
>
> We are deeply grateful to Reviewer QNBN for the constructive comments and feedback. We believe our response have satisfactorily addressed your concerns and clarify our innovations and contributions. Thank you for your valuable feedback and consideration.
>
> **Q1: Sensitivity analysis**
>
> **A1: We analyzed the sensitivity of the penalty coefficient $\beta$.** We emphasize that the selection of the decay coefficient and decay rate is not strict. Referring to Algorithm 1, the decay coefficient only represents the weight of the reconstruction loss in the first iteration. At the beginning of the iteration, minimizing the reconstruction loss is the most important. Therefore, the decay coefficient (the initial weight of the reconstruction loss) is set to 1. The decay rate is matched with the number of iterations. Most settings are feasible as long as they ensure that the decay coefficient approaches 0 at the end of iterations. Although our setting is only one of the feasible settings, it supports all experiments in this paper and is therefore of reference value. On $N=200$ time series with length $T=50$ and number of variables $d=50$, we conducted a sensitivity analysis on the penalty coefficient $\beta$.
>
> |0.001|TPR|F1|SHD|
> |-|-|-|-|
> $W_1$|86.07|66.46|99.0|
> $W_{25}$|92.87|88.37|24.0|
> $W_{50}$|81.48|76.20|54.0|
>
> |0.005|TPR|F1|SHD|
> |-|-|-|-|
> $W_1$|88.91|77.36|51.5|
> $W_{25}$|95.78|94.77|10.0|
> $W_{50}$|87.91|74.52|59.5|
>
> |0.01|TPR|F1|SHD|
> |-|-|-|-|
> $W_1$|77.09|70.02|70.0|
> $W_{25}$|98.95|98.43|3.0|
> $W_{50}$|79.65|77.36|59.5|
>
> |0.05|TPR|F1|SHD|
> |-|-|-|-|
> $W_1$|93.65|95.49|9.0|
> $W_{25}$|98.93|99.46|1.0|
> $W_{50}$|95.09|96.06|8.0|
>
> |0.1|TPR|F1|SHD|
> |-|-|-|-|
> $W_1$|55.95|63.68|55.0|
> $W_{25}$|79.06|86.83|24.0
> $W_{50}$|48.66|59.08|63.0|
>
> The results show that both too small and too large $\beta$ can lead to performance degradation, and DyCausal performs best when $\beta=0.05$, which is consistent with our experimental setting.
>
> **Q2 and W1: Non-smooth dynamic**
>
> **A2: The causal relationship between windows can be non-smooth and largely different.** In this paper, we assume that causal relationships change smoothly over time and use sliding Windows and coarse-grained approximations to learn dynamic causal relationships. However, we emphasize that DyCausal can adapt to non-smooth changing causal relationships. We discuss DyCausal’s behavior in two cases. We set the causal matrix as $W_t[i,j]=W_0[i,j]cos(t/P\pi/E)$, where $P$ regulates the frequency of weight changes, and $E$ regulates the intensity of weight changes. In one case, we set $P=4$, $E=2$, window size $K=3$ and sliding step size $S=1$. On $N=200$ time series with length $T=10$ and number of variables $d=50$, we valid the performance of DyCausal against non-smooth changes. The results are as follows:
>
> | |TPR |F1|SHD|
> |-|-|-|-|
> $W_1$|93.11|92.78|13.7|
> $W_4$|87.92|97.21|5.3|
> $W_8$|91.21|90.93|17.2|
>
> It can be seen that DyCausal accurately learned the causal graphs at both ends and in the middle of the time series. However, we find that the causality within the sliding window do not change at both ends and in the middle of the time series, which might be the reason why DyCausal accurately identified the causal graphs. Suppose the causality changes from $W$ to $W'$ within the sliding window, where the sample size of the causality $W$ is $a$ and the sample size of the causality $W'$ is $b$. By observing the results, we find that the causality learned by DyCausal within this window is $(aW+bW')/(a+b)$. We define the window with changing causality as a transition window; otherwise, it is a stationary window. DyCausal can identify the changing trend of the causality and accurately identify the causal graphs at the stationary window.
>
> In the other case, we set $P=1$, $E=2$, window size $K=1$ and sliding step size $S=1$. The results are as follows:
>
> | |$W_1$|$W_2$|$W_3$|$W_4$|$W_5$|$W_6$|$W_7$|$W_8$|$W_9$|$W_10$|
> |-|-|-|-|-|-|-|-|-|-|-|
> TPR|91.92|97.06|86.87|94.12|86.87|91.18|87.88|90.20|89.90|95.10|
> F1|88.78|92.52|82.30|90.57|83.90|88.15|85.29|88.46|84.76|89.81|
> SHD|23.00|16.00|37.00|20.00|33.00|25.00|30.00|24.00|32.00|22.00|
>
> To deal with frequent and non-smooth changes in the causality, we set the window size to 1. Due to the small sample size within the window of size 1, DyCausal achieved reduced performance but still identified causal graphs with acceptable accuracy at each time step. Although the coarse-grained approximation no longer works when setting window size to 1, it is not usual to change the causality as frequently and strongly as in this experiment. More often, we can set the window size $K>1$ as in the first case, identify the changing trend of the causality, and learn the accurate causal graphs at the stationary window.

---

> ### Author Response · Authors · 2025-11-19
>
> **Q3: Baseline JRNGC**
>
> **A3: We supplement the results of JRNGC on CausalTime.**
>
> |traffic|precision|F1|AUROC|AUPRC|
> |-|-|-|-|-|
> JRNGC|69.39|51.91|73.22|59.97|
> DyCausal|85.00|55.74|78.05|61.43|
>
> |AQI|precision|F1|AUROC|AUPRC|
> |-|-|-|-|-|
> JRNGC|72.46|57.97|83.35|71.71|
> DyCausal|73.81|61.39|73.49|63.89|
>
> |traffic|precision|F1|AUROC|AUPRC|
> |-|-|-|-|-|
> JRNGC|54.41|51.21|61.13|55.62|
> DyCausal|85.51|53.15|70.69|66.06|
>
> We report not only Precision and F1, but also AUROC and AUPRC. The results show that DyCausal outperforms JRNGC in most cases. JRNGC is slightly superior to DyCausal on the AQI dataset.
>
> **Q4: Other real datasets with large-scale nodes**
>
> **A4	: We had provided the results of DyCausal on real datasets with large-scale nodes.** In Appendix C.11. we conducted experiments on real-world NetSim, which records the expression levels of 100 genes and has a verifiable real causal graph. The results on DREAM prove that DyCausal outperforms the comparison methods on large-scale real datasets. Unfortunately, we did not find any other larger-scale datasets that contain verifiable real causal graphs. However, we are grateful to the reviewer YHJh for providing the benchmark CausalRivers, which extensively records river flow, assumes fixed causal structures with dynamic edge weights and is highly suitable for verifying the dynamic causal discovery performance of DyCausal. We conduct experiments on the datasets with five nodes provided by CausalRivers.
>
> | |AUROC|
> |-|-|
> DyCausal|0.7142|
> CUTS+|0.6072|
> NGM|0.5357|
>
> The results show that DyCausal outperforms the comparison methods. This may be because DyCausal is able to capture the weights of the dynamics.
>
> **Q5: How do static methods get results?**
>
> **A5: We directly applied DYNOTEARS and NTS-NOTEARS to dynamic datasets and forced the learning of static causal relationships.** The purpose of this protocol is to demonstrate the predicament of static methods when learning dynamic causality. From the results, static methods are almost unable to accurately learn dynamic causal relationships.
>
> **W1: Non-smooth and linear assumptions**
>
> **A6: DyCausal can handle non-smooth and nonlinear causal relationships.** See our answer A2, DyCausal can handle the non-smooth causality and accurately identify causal graphs. In addition, Eq.(8) and (9) and experiments had shown that DyCausal is not only for the linear causaliy, but also flexibly learn various causal relationships.
>
> **W2: Whether the acyclic constraint is innovative?**
>
> **A7: Our improvement to the acyclic constraint is novel and crucial.** As shown in Appendix C.10, the acyclic constraints of DAGMA may suffer catastrophic failures. We have explained the reason for the failure, that is, $h_{log}$ requires the constraint to be optimized in a limited space, once the constraint exceeds the optimizable space, it is forced to scale into the optimizable space (the scaling strategy is to roll back the matrix before optimization), and re-optimize. Our $h_{norm}$ is equivalent to scaling before optimization (the scaling strategy is to divide the matrix by its 1-norm). We have demonstrated that $h_{norm}$ inherits the excellent properties of $h_{log}$ and satisfies the stability criteria, while $h_{log}$ cannot meet all the stability criteria. For a fair comparison, we directly replace DAGMA's $h_{log}$ with $h_{norm}$ and conduct experiments on non-temporal data.
>
> | |TPR|FDR|SHD|
> |-|-|-|-|
> $h_{norm}$|84.10|6.82|22.4|
> $h_{log}$|78.90|11.80|29.7|
>
> The results show that the performance of $h_{norm}$ is superior to that of $h_{log}$.
>
> **W3: Lack theoretical support**
>
> **A8: DyCausal is guaranteed to be identifiable.** No matter it is directly encoding the causal graph or extracting the causal graph from the model parameters, the identifiability of the causal graph is guaranteed by the score function. The score function typically consist of a reconstruction loss and a regularization term, while causal graphs is a DAG that can minimize the score function. This is the essence of DyCausal encodes the causal graph through the model, or traditional methods like NTS-NOTEARS extract the causal graph from the model parameters. From this perspective, the score function equally guarantees the identifiability of the causal graph under DyCausal and traditional methods such as NTS-NOTEARS. We have further analyzed the identifiability of dynamic causality in Appendix D. Specifically, the identifiability of dynamic causal relationships can be broken down into the identifiability of approximate static causal relationships within multiple short time Windows. When the observed data is sufficient and the causal structures encoded by the model can minimize the optimization objective (that is, the causal structure is a DAG and minimizes the score function), the approximate static causal relationships within a short time window are identifiable, and thus the dynamic causal relationships across the entire time series are also identifiable.

---

> > ### Comment · Reviewer_QNBN · 2025-11-25
> >
> > Thanks a lot for your detailed reply, which I think addressed most of my concerns. I have adjusted my rating accordingly.
> >
> > **About the large size of the benchmark**: I agree with the authors that $d=100$ datasets are already fairly large. We also see that the authors incorporated a real-world dataset on **CausalRivers** during the rebuttal phase.
> >
> > **Thanks for clarifying the innovation of the acyclic constraint $h_{norm}$**. I have clarified the difference between the constraints you proposed and DAGMA, but I think more examples of visualization on time series are needed to further clarify your advantages.
> >
> > I'm still confused about what to do if we make the causal matrix of both the intra-slice and inter-slice time-varying. If the timestep is 7 and lag is 1, shouldn't we expect $7 \times 2 = 14$ causal graphs? Or are you getting a summary graph? If we can get answers to these two questions, we will consider further increasing the score.
> >
> > I hope the author will incorporate these discussions into the revision.

---

> ### Author Response · Authors · 2025-11-27
>
> Thank you once again for your valuable feedback and positive rating. We are delighted to know that the previous response has addressed most of your concerns. Below, we address your remaining concerns.
>
> **Q1: Visualization of $h_{norm}$**
>
> **A1:**
>
> 1. **We had included some visualization results in the paper to prove that $h_{norm}$ outperforms $h_{log}$.** In Figure 4, we plotted the curves of various acyclic constraints with the matrix weight and the cycle length. It can be seen that $h_{norm}$ does not vanishes as the weight decreases and the cycle length increases, nor does it grow to infinity as the weight increases. In contrast, $h_{log}$ does not have these excellent properties. This proves that $h_{norm}$ is more stable and effective than $h_{log}$. Figures A9 and A10 recorded a catastrophic failure of $h_{log}$. It can be seen that $h_{log}$ frequently exceeds the optimizable space and is forced to roll back, which not only leads to more iterations but also fails the optimization of $h_{log}$.
> 2. **We further compare DyCausal and DyCausal with $h_{log}$ on the causal graph with 10 nodes and dynamic settings, where the maximum lag is 1.** The additional visualization results, Figures A11 and A12 show the estimated causal graphs and the trajectory of $h_{norm}$ and $h_{log}$ when $h_{log}$ runs successfully. The results show that the use of $h_{norm}$ can identify more accurate causal structures than that of $h_{log}$. Especially at both ends of the time series ($W_1$ and $W_{50}$), DyCausal almost perfectly identifies the causal graphs, while DyCausal with $h_{log}$ struggles with redundant and missing edges. The trajectories of $h_{norm}$ and $h_{log}$ indicate that $h_{norm}$ decreases more steadily early in training and is closer to 0 at the end of training.
>
> **Q2: time-varying intra-slice and inter-slice causal matrices.**
>
> **A2: DyCausal can handle time-varying causality in both intra-slice and inter-slice matrices.** We had conducted experiments in Section 5.1 to show the excellent performance of DyCausal in learning the intra-slice and inter-slice causal matrices. At t=0, we cannot obtain the previous observations, and naturally cannot obtain lagged and instantaneous causality. Therefore, if the time step is 7 and the maximum lag is 1, we set the time series length to 8 and expect 7×2=14 causal graphs. We will clarify this description in the article. We conduct DyCausal on N=200 time series of length 11 (time step is 10) and the number of variables d=20, and report the results of instantaneous and lagged causal relationships at each time step (a total of 10×2=20 causal matrices).
>
> |instantaneous|$W_1$|$W_2$|$W_3$|$W_4$|$W_5$|$W_6$|$W_7$|$W_8$|$W_9$|$W_{10}$|
> |-|-|-|-|-|-|-|-|-|-|-|
> TPR|86.91|83.03|86.12|81.39|91.29|91.29|81.56|85.19|83.60|83.64
> F1|89.27|89.53|91.80|89.43|93.62|93.88|89.03|90.19|89.59|86.47
> SHD|6.2|9.8|8.5|8.7|2.7|2.8|8.8|9.5|10.3|8.0
>
> |lagged|$W_1$|$W_2$|$W_3$|$W_4$|$W_5$|$W_6$|$W_7$|$W_8$|$W_9$|$W_{10}$|
> |-|-|-|-|-|-|-|-|-|-|-|
> TPR|90.81|86.14|90.85|84.25|93.16|93.13|84.02|87.56|86.67|90.28
> F1|93.50|91.57|94.79|91.12|95.75|96.32|90.50|92.29|91.84|92.43
> SHD|4.8|8.4|5.9|7.5|2.5|2.1|8.3|8.7|8.3|5.7
>
> The results show that DyCausal accurately identifies the instantaneous and lagged causal relationships.
>
> We believe that our response has fully addressed your concerns. We are incorporating our discussions into the revision to the best of our ability, and believe these discussions will significantly improve the paper.

---

> > ### Comment · Reviewer_QNBN · 2025-11-27
> >
> > **I thank the reviewers for their extensive rebuttal. Most of my concerns were adequately addressed.**
> >
> > I hope the authors can incorporate the above discussion into the paper's revision, especially the section on Causal Rivers, which I've noticed is also of interest to other reviewers (such as YHjh).

---

> > > ### Author Response · Authors · 2025-11-27
> > >
> > > Thank you for your valuable feedback and positive assessment. We have incorporated the crucial issues and experimental results from our discussion into the revision. We believe that these supplements have clarified the superior performance of DyCausal and significantly enhanced our confidence in the results.

---

### Official Review · Reviewer_DRWo · 2025-10-30

**Soundness:** 2
**Presentation:** 2
**Contribution:** 1
**Rating:** 2
**Confidence:** 4

**Summary:**

The paper proposes DyCausal, a framework for learning dynamic causal structures in time series that exhibit complex and time-varying dependencies commonly observed in real-world systems. The method addresses fully dynamic causality, where both instantaneous and lagged dependencies evolve over time. To tackle this challenge, DyCausal leverages convolutional networks to capture causal patterns within coarse-grained time windows, and applies linear interpolation to refine the causal structures at each time step, thereby recovering fine-grained and continuously evolving causal graphs. In addition, an acyclicity constraint based on matrix norm scaling is introduced to enhance computational efficiency while effectively preventing loops in the evolving causal structures. Experimental results demonstrate the effectiveness of the proposed approach.

**Strengths:**

1. The paper extends causal structure learning to a general setting of fully dynamic causality, where both instantaneous and lagged dependencies evolve over time. This broadens the applicability of causal discovery methods to more realistic time-varying systems.

2. The proposed approach appears technically sound.

**Weaknesses:**

1. The paper lacks significant technical innovation. Most components of the proposed framework appear to be incremental adaptations of existing techniques rather than conceptually new contributions. The use of convolutional networks for coarse-grained temporal modeling and the acyclicity constraint are engineering extensions that do not substantially advance the methodological frontier. Moreover, the theoretical results are mostly straightforward derivations of [1], without introducing new analytical insights.

2. The presentation and notation require improvement. For example, in Theorem 2 the symbol $=\in$ is incorrect; in Algorithm 1, the L₁-norm notation is inconsistent; and the usage of boldface for matrices and vectors is not uniform (e.g., the identity matrix I on line 214). In addition, the definition of Δ in Theorem 1 (line 239) is unclear, and line 7 of Algorithm 1 is difficult to interpret. These issues hinder clarity.

3. The experimental section contains several marking or annotation errors. For instance, on line 1098, the reported TPR: NTS-NO better than DyCausal. Similarly, see line 832.

[1] Bello K, Aragam B, Ravikumar P. Dagma: Learning dags via m-matrices and a log-determinant acyclicity characterization[J]. Advances in Neural Information Processing Systems, 2022, 35: 8226-8239.

**Questions:**

1. What are the limitations introduced by the coarse-grained approximation?

2. Can the authors establish theoretical guarantees for convergence of proposed framework, similar to the Lemma 6 in [1]?

---

> ### Author Response · Authors · 2025-11-19
>
> We sincerely appreciate the reviewer DRWo for their comments. We hope that our response have adequately address your concerns and misunderstandings and clarify the contributions of our work. Thank you for your valuable feedback and consideration.
>
> **Q1: Limitations of coarse-grained approximation**
>
> **A1: The coarse-grained approximation is bound to introduces some limitations.** Specifically, the matrix approximated through linear interpolation may not be the same as the real matrix, which depends on the sliding step size of the window. The smaller the step size, the smaller the interpolation interval, and the closer the approximate matrix is to the real matrix. In addition, when the true causality changes non-smoothly, coarse-grained approximation may not be applicable. However, our DyCausal can naturally handle the non-smooth changing causality. It only needs to set the sliding window size and sliding step size to 1, which is equivalent to directly learning fine-grained causal relationships.
>
> **Q2: Theoretical guarantees of convergence**
>
> **A2: It is easy to prove the convergence similar to the Lemma 6 in [1].** We define $\frac{W\circ W}{||W\circ W||_ 1}=A$, and the acyclic constraint $h_{norm}(A)$ is equivalent to $h_{log}(A)=-\log\det (\alpha I-A) +d\log\alpha$, $\alpha=1$. According to DAGMA's Corollary 3, since $\alpha=1>0$ and $\rho(A)=\rho(\frac{W\circ W}{||W\circ W||_ 1})<1$, $h_{log}(A)$ is a convex function, $h_{norm}(A)=h_{log}(A)$ is also a convex function and all its stations are global minimum values, which correspond to DAGs. Therefore, for our algorithm 1, when $\mu=0$, we solve the following optimization problem: $\hat{A}=\arg\min_{\hat{A}}h_{norm}(\hat{A})$. Due to the convex function property of $h_{norm}(A)$, the solution $\hat{A}$ must be DAG. Finally, there is an obvious conclusion: if $\hat{A}$ is a DAG, then whether it is $\hat{A}=\hat{W}\circ\hat{W}$ for $h_{log}$ or $\hat{A}=\frac{\hat{W}\circ\hat{W}}{||\hat{W}\circ\hat{W}||_ 1}$ for $h_{norm}$, $\hat{W}$ is a DAG. In this way, we prove the convergence guarantee similar to Lemma 6 in [1].
>
> **W1: Lacks significant technical innovation**
>
> **A3: DyCausal is novel and theoretically guaranteed. You may seriously misunderstand our algorithm.** DyCausal is not a combination of existing technologies, but a new approach to learning fully dynamic causal structures. Before this, how to learn fully dynamic causal structures has not been fully explored. Specifically, DyCausal uses convolutional networks and parallel encoding networks to encode the causal matrices $W_t$, which is different from existing methods of directly extracting $W$ from the model parameters. In addition, we introduce a coarse-grained approximation strategy of dynamic causal structure, which is not forcibly added but naturally fits with the sliding window. The extensive experimental results demonstrate the superiority of the above causal structure learning strategies. Moreover, our improvement of the acyclic constraint is crucial. As shown in Appendix C.10, the improved acyclic constraint addresses an undeniable defect of $h_{log}$, which can only be optimized in a limited space. Furthermore, we demonstrate that $h_{norm}$ conforms to the stability criterion, while $h_{log}$ has previously been proven to be unstable. For a fair comparison, we also directly replace DAGMA's $h_{log}$ with $h_{norm}$ and conduct experiments on non-temporal data with the sample size $N=1000$ and the number of variables $d=50$.
>
> | |TPR|FDR|SHD|
> |-|-|-|-|
> $h_{norm}$|84.10|6.82|22.4|
> $h_{log}$|78.90|11.80|29.7|
>
> The results show that the performance of $h_{norm}$ is superior to that of $h_{log}$.
>
> **W2: Annotation errors**
>
> **A4: There are some factual errors in your understanding of our notations.** We have fixed the wrong notations and explained the meaning of some notations to facilitate understandings. **But we must clarify some notations to prevent you from misinterpreting them as errors.** In Theorem 2, we represent $\frac{W\circ W}{||W\circ W||_ 1}$ as $A$ to simplify the symbol, which is to make the stability criterions easier to read. The meaning of $\nabla$ is obvious, that is, the gradient of $h_{norm}$. We deliberately distinguish $|W|_1$ and $||W||_1$ in Algorithm 1. In our paper, $|W|_1$ is the sum of the absolute values of each term in the matrix, and it is a regularization term. $||W||_1$ is the matrix 1-norm, which is the maximum sum of the absolute values of column vectors in the matrix. We had explained the meaning of line 7 in Algorithm 1, that is, pruning the edges with weights smaller than $\delta$ to obtain the causal graphs.

---

> > ### Author Response · Authors · 2025-11-27
> >
> > We sincerely appreciate your valuable feedback and consideration. We have read your suggestions and addressed your concerns point by point. We are looking forward to more discussions with you and believe that this will address your other concerns and improve your assessment of our work.

---

### Official Review · Reviewer_wtX2 · 2025-11-01

**Soundness:** 3
**Presentation:** 3
**Contribution:** 3
**Rating:** 6
**Confidence:** 3

**Summary:**

This paper proposes structure learning for dynamic (i.e., time-varying) causal graphs in time series. This is proposed via a "coarse-to-fine" strategy; this proceeds by roughly identifying causal structures over some large sliding window, then linearly interpolating. An additional modification to the acyclicity penalty of DAGMA is introduced. Extensive experiments are undertaken, which show superior performance of the proposed method in real and synthetic datasets with time-varying causal mechanisms.

**Strengths:**

- The paper is well-written and easy to follow.
- The paper motivates and undertakes an important and under-discussed problem of time-varying causal structure learning.
- Relative to some prior work (e.g., DyCast), which only learns time-varying instantaneous graphs, DyCausal simultaneously learns time-varying lagged relationships. This seems to significantly improve performance.
- The experiments clearly demonstrate strong performance relative to relatively recent and strong baselines.

**Weaknesses:**

I'm not totally convinced by the analysis of the revised log-determinant penalty. One reason why is that the benefits of DAGMA lie partially in nice optimization properties in a barrier method approach -- it's a priori possible but not obvious to me that the proposed version would perform better in the general structure learning setting. The ablation in C.10 is nice in theory, but the graph in Figure A9 seems to moreso suggest some catastrophic failure than poor optimization.

**Questions:**

1. How does h_log compare with more extensive hyperparameter tuning? Do the results of C.10 change? Furthermore, does the proposed version h_norm significantly change results in the general (static) structure learning setting?

2. How is initialization performed? Dagma initializes W to be the zero matrix, but this obviously results in an infinite denominator.

---

> ### Author Response · Authors · 2025-11-19
>
> We sincerely thank Reviewer wtX2 for their valuable comments and positive rating. We believe our response can address your concerns.
>
> **Q1:$h_{norm} vs. h_{log}$**
>
> **A1: $h_{norm}$ is significantly superior to $h_{log}$**. Parameter tuning does not improve the performance of $h_{log}$. As you said, what we show in Figure A9 is the catastrophic failure of $h_{log}$. We have already explained the reason for the failure, that is, $h_{log}$ requires optimizing the constraint term within a limited space. Once the constraint term exceeds the optimizable space, it is forced to scale into the optimizable space (the scaling strategy is to roll back the matrix) and re-optimize. Our $h_{norm}$ is equivalent to scaling before optimization (the scaling strategy is to divide the matrix by its 1-norm), and we have demonstrated that $h_{norm}$ not only inherits the excellent properties of $h_{log}$ but also meets the stability criteria, while $h_{log}$ failed to satisfy all stability criteria. For a fair comparison, we directly replace DAGMA's $h_{log}$ with $h_{norm}$ and conduct experiments on non-temporal data with the sample size $N=1000$ and the number of variables $d=50$.
>
> | |TPR|FDR|SHD|
> |-|-|-|-|
> $h_{norm}$|84.10|6.82|22.4|
> $h_{log}$|78.90|11.80|29.7|
>
> The results show that the performance of $h_{norm}$ is superior to that of $h_{log}$. In addition, we found that $h_{log}$ still exceeds the optimizable space on non-temporal data and forces the model to fall back. The results in Appendix C.10 had shown that this defect of $h_{log}$ may lead to catastrophic accuracy and additional running time. In conclusion, $h_{norm}$ is a superior of $h_{log}$.
>
> **Q2: How to initialize $W$**
>
> **A2: Our acyclic constraints will not be infinite.** We have reported the variation curve of $h_{norm}(kA)$ with k in Figure 4. When k approaches 0, $h_{norm}(kA)$ does not increase infinitely. This is because the 1-norm is the maximum sum of the absolute values of the column vectors in the matrix. Therefore, for matrix $\frac{W\circ W}{||W\circ W||_1}$, the denominator is always larger than (or equal to) the numerator. Secondly, you might misunderstand DyCausal. Our matrix $W_t$ is encoded by the neural network through Eq.(3) and Eq.(4), rather than being extracted from the model parameters, so there is no initialization process. In addition, the parameters of the encoding network are obtained through random initialization. All of the above ensure that the acyclic constraint cannot be infinite.

---

> > ### Author Response · Authors · 2025-11-27
> >
> > Thank you sincerely for your valuable comments and positive evaluations. We have addressed your concerns point by point. Furthermore, in Appendix C.10, we have supplemented the comparison of $h_{norm}$ and $h_{log}$under dynamic settings when $h_{log}$runs successfully. The results further proved the superiority of $h_{norm}$. If you still have other concerns, we are looking forward to addressing them to further improve your assessment of our work.

---

### Author Response · Authors · 2025-11-21

We sincerely thank all reviewers for their valuable comments and thoughtful feedback, which has been invaluable to us in improving this work. We are particularly grateful for your recognition of DyCausal's contributions and effectiveness in addressing dynamic causal discovery.

We have adjusted the article based on the reviewers' comments and added more detailed explanations for the content that may have been misunderstood.

For the specific questions and suggestions from each reviewer, we provide individual point-by-point responses in their respective sections below. We believe that these clarifications and additional results adequately address the raised concerns and further strengthen our contributions in novelty and theory.

---

### Author Response · Authors · 2025-11-27

Thank you again to all the reviewers for your valuable comments and thoughtful feedback, which has been invaluable in helping us improve this work. We have incorporated the main issues and experimental results from our discussion into the revision, which will further clarify the superior performance of DyCausal and strengthen our confidence in the results.

---

### Author Response · Authors · 2025-11-29
**Summary of the discussion for the Area Chair**

Thank you to all the reviewers for your considerable time and effort in the review, and for your positive assessment. We have been informed that all review scores have been reverted to their pre-discussion status, and no further reviewer discussions or public comments are allowed. We sincerely appreciate the Area Chair’s time and effort in considering our responses and discussions, and his or her fair review. To facilitate the area chair to consider the existing discussion and improved scores, **we summarize the discussions with all the reviewers:**

$\mathbf{Score:}$

By the time the review scores were reverted, we have received the following ratings:

**Reviewer wtX2: 6**

**Reviewer DRWo: 2**

**Reviewer QNBN: 8**

**Reviewer YHJh: 6**

Most reviewers with a positive assessment. We have extensive discussions with the Reviewers QNBN and YHJh, and fully addressed their concerns. Reviewers QNBN and YHJh have approved the innovation, robustness and contribution of this paper, where **Reviewer QNBN has improved the rating to 8 and Reviewer YHJh maintains a positive rating.**

$\mathbf{For\ the\ Reviewer\ wtX2\ (rating: 6):}$

**The Reviewer wtX2 has approved the outstanding contribution and superior performance of this paper in dynamic causal discovery.** In rebuttal, we have demonstrated that $h_{norm}$ is significantly superior to $h_{log}$ through a fair comparison on non-temporal data. In addition, we have clarified how to initialize the causal matrix $W$, which will not be infinite. The above responses have addressed the concerns of the Reviewer wtX2.

$\mathbf{For\ the\ Reviewer\ DRWo\ (rating: 2):}$

**We must clarify that the Reviewer DRWo has serious misunderstandings and biases regarding the technical innovation.** Other reviewers have clearly approved the technical innovation, robustness and contribution of this paper. Specifically, we propose DyCausal to learn fully dynamic causal relationships by innovatively combining convolutional networks and coarse-grained approximations. Extensive experiments have demonstrated the superiority and robustness of DyCausal. We also innovatively propose a globally optimizable acyclic constraint that satisfies the stability criterions. In addition, we have addressed the concerns of the reviewer DRWo about the limitations of coarse-grained approximations and the guarantees of convergence, and clarified his or her misunderstandings and factual errors about notation.

$\mathbf{For\ the\ Reviewer\ QNBN\ (rating: 8):}$

**The Reviewer QNBN has affirmed the innovation, robustness and contribution of this paper and raised the rating to 8.** We are delighted to have an in-depth discussion with the Reviewer QNBN and fully address his or her concerns. Specifically, we have analyzed the sensitivity of the penalty coefficient $\beta$, clarified the robustness of DyCausal to non-smooth dynamic causal relationships, the innovation of the proposed acyclic constraint, and the identifiability of dynamic causality. We have also provided comparison results with the state-of-the-art baseline JRNGC and visualization results of $h_{norm}$.

$\mathbf{For\ the\ Reviewer\ YHJh\ (rating: 6):}$

**The Reviewer YHJh has approved the innovation and robustness of this paper, especially the contribution in leading the direction of future work, and maintains a positive assessment.** Through extensive discussions, the concerns of the Reviewer YHJh have been fully addressed by us. Specifically, we have clarified the robustness of DyCausal to non-smooth dynamic causality, inter-slice/intra-slice causality, and cyclic causality, and demonstrated the technical innovation and importance of convolutional windows, coarse-grained approximations, and the proposed acyclic constraint. We have also provided the superior results of DyCausal on the real dataset CausalRivers.

We have incorporated the main issues and experimental results from our discussion into the revision. We promise that the summaries of the review scores and discussions are all true and reliable. Thank you once again to the Area Chair for your efforts to provide a fair review.

---

### Meta-Review · Area_Chair_K8gD · 2026-01-05

**Summary:**

This paper proposes DyCausal, a novel framework for learning fully dynamic causal graphs in time-series data, where both instantaneous and lagged causal relationships evolve over time. DyCausal adopts a coarse-to-fine strategy, using convolutional networks to learn causal structures within sliding time windows and linear interpolation to recover fine-grained temporal dynamics. A key technical contribution is a globally optimizable and stable acyclicity constraint, improving upon DAGMA by avoiding catastrophic optimization failures. Extensive experiments on synthetic, semi-synthetic, and real-world datasets (including CausalTime, NetSim, CausalRivers, and DREAM benchmarks) demonstrate DyCausal’s robustness, superior performance, and ability to handle non-smooth and cyclic causal dynamics.

**Reviewer Concerns:**

Reviewer wtX2 questioned whether the revised acyclicity constraint truly improves upon DAGMA under broader hyperparameter tuning and asked how the causal matrix is initialized without causing numerical instability. Reviewer DRWo argued that DyCausal lacks significant technical innovation, raised concerns about notation and presentation errors, and questioned the theoretical guarantees and limitations of the coarse-grained approximation. Reviewer QNBN was concerned about strong smoothness assumptions, lack of sensitivity analysis, missing state-of-the-art baselines (e.g., JRNGC), identifiability of dynamic causality, and ambiguity in how time-varying intra-slice and inter-slice causal graphs are defined and evaluated. Reviewer YHJh asked for clarification on evaluation protocols for dynamic graphs, justification of real-world datasets with dynamic causality, enforcement of acyclicity, granular reporting of instantaneous versus lagged links, and stronger validation on benchmarks such as CausalRivers.

**Reviewer Scores:**

After going through the discussion between authors and reviewers, I think most of the problems are addressed. And the reviewers would have raised their scores.

---

### Decision · Program_Chairs · 2026-01-26

Accept (Poster)